# Determinants of RNA metabolism in the *Schizosaccharomyces pombe* genome

Philipp Eser[1,2,†], Leonhard Wachutka[2,‡,†], Kerstin C Maier[1], Carina Demel[1], Mariana Boroni[2], Srignanakshi Iyer[2], Patrick Cramer[1,*] & Julien Gagneur[2,‡,**]

## Abstract

To decrypt the regulatory code of the genome, sequence elements must be defined that determine the kinetics of RNA metabolism and thus gene expression. Here, we attempt such decryption in an eukaryotic model organism, the fission yeast *S. pombe*. We first derive an improved genome annotation that redefines borders of 36% of expressed mRNAs and adds 487 non-coding RNAs (ncRNAs). We then combine RNA labeling *in vivo* with mathematical modeling to obtain rates of RNA synthesis and degradation for 5,484 expressed RNAs and splicing rates for 4,958 introns. We identify functional sequence elements in DNA and RNA that control RNA metabolic rates and quantify the contributions of individual nucleotides to RNA synthesis, splicing, and degradation. Our approach reveals distinct kinetics of mRNA and ncRNA metabolism, separates antisense regulation by transcription interference from RNA interference, and provides a general tool for studying the regulatory code of genomes.

**Keywords** *cis*-regulatory element; gene regulation; RNA degradation; RNA synthesis; splicing

**Subject Categories** Genome-Scale & Integrative Biology; RNA Biology; Chromatin, Epigenetics, Genomics & Functional Genomics

**Mol Syst Biol. (2016) 12: 857**

## Introduction

Gene expression can be regulated at each stage of RNA metabolism, during RNA synthesis, splicing, and degradation. The ratio between the rates of RNA synthesis and degradation determines steady-state levels of mature RNA, thereby controlling the amount of messenger RNA (mRNA) and the cellular concentration of non-coding RNA (ncRNA). The rates of both RNA degradation and splicing contribute to the time required for reaching mature RNA steady-state levels

following transcriptional responses (Jeffares *et al*, 2008; Rabani *et al*, 2014).

To estimate the kinetics of RNA metabolic events genome-wide, techniques including genomic run-on followed by RNA polymerase chromatin immunoprecipitation (Pelechano *et al*, 2010), cytoplasmic sequestration of RNA polymerase (Geisberg *et al*, Cell, 2014), or metabolic RNA labeling (Miller *et al*, 2011; Rabani *et al*, 2011; Zeisel *et al*, 2011; Schulz *et al*, 2013) have been performed in various organisms and under different conditions. Quantifying the individual contributions of synthesis and degradation led to an improved understanding of how these processes are coordinated and how they control mRNA levels. The rates of RNA synthesis show large variation across genes and are the major determinants of constitutive and temporally or conditionally changing mRNA levels (Schwanhäusser *et al*, 2011; Marguerat *et al*, 2014; Rabani *et al*, 2014). RNA degradation modulates and fine-tunes mRNA abundance, largely varies across conditions and between organisms, and can be dynamically changed to shape gene expression (Munchel *et al*, 2011; Pai *et al*, 2012; Sun *et al*, 2012; Eser *et al*, 2014).

In contrast to synthesis and degradation rates, accurate genome-wide kinetic parameters of splicing are still lacking, likely because sequencing depth is more limiting to get measurements of short-lived precursor RNAs. Nonetheless, recent studies in human (Windhager *et al*, 2012) and mouse (Rabani *et al*, 2011, 2014) indicate that the rates of splicing also vary within a wide range. However, how these rates are quantitatively encoded in the genome remains largely unknown.

The fission yeast *Schizosaccharomyces pombe* (*S. pombe*) is an attractive model organism to study eukaryotic RNA metabolism. *S. pombe* shares important gene expression mechanisms with higher eukaryotes that are not prominent or even absent in the budding yeast *S. cerevisiae*. These include splicing, which occurs for ~50% of the genes and is achieved with conserved spliceosomal components (Käufer & Potashkin, 2000) and conserved consensus splice site (SS) sequences (Lerner *et al*, 1980; Roca & Krainer, 2009), heterochromatin silencing (Allshire *et al*, 1995), and RNA interference (Volpe *et al*, 2002). Because of its relevance for studying eukaryotic gene expression, *S. pombe* has been extensively characterized by

1 Department of Molecular Biology, Max Planck Institute for Biophysical Chemistry, Göttingen, Germany
2 Gene Center Munich and Department of Biochemistry, Center for Integrated Protein Science CIPSM, Ludwig-Maximilians-Universität München, Munich, Germany
  *Corresponding author. Tel: +49 551 201 2800; E-mail: patrick.cramer@mpibpc.mpg.de
  **Corresponding author. Tel: +49 89 289 19411; E-mail: gagneur@in.tum.de
  †These authors contributed equally to this work
  ‡Present address: Department of Bioinformatics & Computational Biology, Technische Universität München, Garching/Munich, Germany

genomic studies, and this led to an annotation of transcribed loci that includes ncRNAs (Dutrow *et al*, 2008; Wilhelm *et al*, 2008; Rhind *et al*, 2011), a map of polyadenylation sites (Mata, 2013; Schlackow *et al*, 2013), the "translatome" as measured by ribosome profiling (Duncan & Mata, 2014), and an absolute quantification of protein and RNA (Marguerat *et al*, 2012).

Here, we used the fission yeast *S. pombe* as a model system to quantify RNA metabolism genome-wide, to identify genomic regulatory elements at single-nucleotide resolution, and to quantify the contribution of these elements to the kinetics underlying RNA metabolism. We provide an improved genome annotation and a quantitative description of RNA metabolism for an important eukaryotic model organism. The approach developed here enables quantitative, genome-wide studies of eukaryotic gene regulation and provides a general route to help decrypting the regulatory code of the genome.

# Results

## Strategy to describe RNA metabolism and regulatory elements

Our approach consists of three steps (Fig 1). First, we performed short and progressive metabolic labeling of RNA with 4-thiouracil coupled with strand-specific RNA-seq (4tU-Seq, Materials and Methods). With the use of advanced computational modeling, we obtained accurate estimates of RNA synthesis and degradation rates for 5,484 transcribed loci and splicing rates for 4,958 splice sites. Second, a novel statistical modeling procedure quantifies the contribution of each single nucleotide in predicting RNA metabolic rates and thereby identifies sequence features that contribute to RNA metabolism rates. We then supported a causal role of these features by comparing RNA expression fold changes between strains differing by a single nucleotide at these sites with the corresponding fold-changes predicted by the model. Our approach relies on an accurate annotation of the genome. In particular, accurate transcript boundaries are important for quantifying RNA metabolism. We therefore first set out to precisely define the transcriptional units in *S. pombe*.

## Mapping transcriptional units in *S. pombe*

To map transcribed regions in the *S. pombe* genome, we carried out strand-specific, paired-end deep sequencing of total RNA (RNA-seq, at mean per-base read coverage of 385) from fission yeast grown in rich media (Materials and Methods). Genomic intervals of apparently uninterrupted transcription (transcriptional units, TUs, Fig 2A) were identified with a segmentation algorithm applied to the RNA-seq read coverage signal (Materials and Methods). The three parameters of the algorithm, the minimum per-base coverage, the minimum TU length, and the maximum gap within TUs, were chosen to best match the existing genome annotation (Pombase version 2.22 (Wood *et al*, 2012), Appendix Fig S1A and B). TUs that did not show significant signal in the 4tU-Seq dataset were considered as artifacts and discarded (Materials and Methods).

The segmentation led to a total of 5,484 TUs (Fig 2B, Dataset EV1), of which 4,105 were containing a complete, annotated open reading frame (ORF-TU), 1,014 were non-coding TUs (ncTU), and the remaining 365 TUs contained two or more annotated adjacent transcripts. We classified these 365 TUs as potential multicistronic

RNAs. As RNA-seq coverage alone does not allow distinction between overlapping signal of adjacent genes from true multi-cistronic loci, it should remain clear that many of them might contain independently transcribed regions. Only a small number of novel splice sites were identified (148 out of 4,958, Dataset EV2), and no evidence for substantial alternative splicing at any given intron or circular RNAs was found (at least 10 supporting reads, Materials and Methods). These observations are in line with previous RNA-seq studies of *S. pombe* showing that alternative splicing is prevalent but rare (Rhind *et al*, 2011; Bitton *et al*, 2015). A total of 402 ORFs (8%) in the existing annotation (Wood *et al*, 2012) were not recovered (Fig 2C, Materials and Methods), apparently because they were not expressed under the used growth condition (gene set enrichment analysis, Appendix Fig S1C).

## Improved annotation of transcribed regions in *S. pombe*

The resulting annotation of ncTUs in *S. pombe* differed largely from the current one. We identified 487 novel ncTUs, changed the boundaries by more than 200 nt of 422 (27%) previously annotated ncRNAs and could not recover 1,011 (66%) of the previously annotated ncRNAs (Materials and Methods, Fig 2B and C). A large fraction of the latter apparently represent spurious antisense RNAs that are often generated with conventional protocols, but their generation was suppressed here with the use of actinomycin D (Perocchi *et al*, 2007). Indeed, 49% of those non-recovered ncRNAs were located antisense to highly expressed ORF-TUs and showed on average 66-fold higher antisense than sense coverage (Appendix Fig S1D). The remaining half non-recovered RNAs might be genuine ncRNAs that are not expressed in our growth condition. Thus, we redefined the location and boundaries of most ncRNAs in *S. pombe*, leaving only 105 of the currently annotated ncRNAs unchanged.

We also redefined boundaries for 1,481 coding transcripts that differed from the existing annotation by at least 200 nt. Untranslated regions (UTRs) of ORF-TUs were generally much shorter than previously annotated (mean difference 91 nt), consistent with a previously curated set of ORF transcript boundaries (Lantermann *et al*, 2010). This difference apparently also stemmed from spurious antisense RNAs in previous datasets because 68% of the 376 3′ UTRs that were at least 250 nt shorter in our annotation showed higher antisense than sense coverage (Appendix Fig S1E; for an example see Fig 2A). Our transcript 3′-ends were centered around experimentally mapped polyadenylation (polyA) sites (Mata, 2013), whereas the previously annotated 3′-ends typically extended well beyond polyA sites (median difference = 3 nt versus 45 nt, Fig 2D). Similarly, our transcript 5′-ends were centered around experimentally mapped transcription start sites (Li *et al*, 2015), whereas previously annotated 5′-ends extended beyond (median difference = 0 nt versus 44 nt, Fig 2D). Thus, our map of TUs provides significant changes to the annotation of the *S. pombe* genome, removing false-positive ncRNAs from the current annotation and shortening aberrantly long UTRs.

## Quantification of *S. pombe* RNA metabolism

To quantify the kinetics of RNA synthesis, splicing, and degradation genome-wide, we sequenced newly synthesized RNA after metabolic RNA labeling with 4-thiouracil (4tU-Seq) and used the obtained data for kinetic modeling (Fig 1, step 1). We used 4tU instead of the more

## 1. Genome-wide *in vivo* RNA metabolism kinetics

- 4tU RNA labeling and deep sequencing in fission yeast

- Mathematical modeling

- RNA metabolism rates for 5,484 genes and 4,958 introns

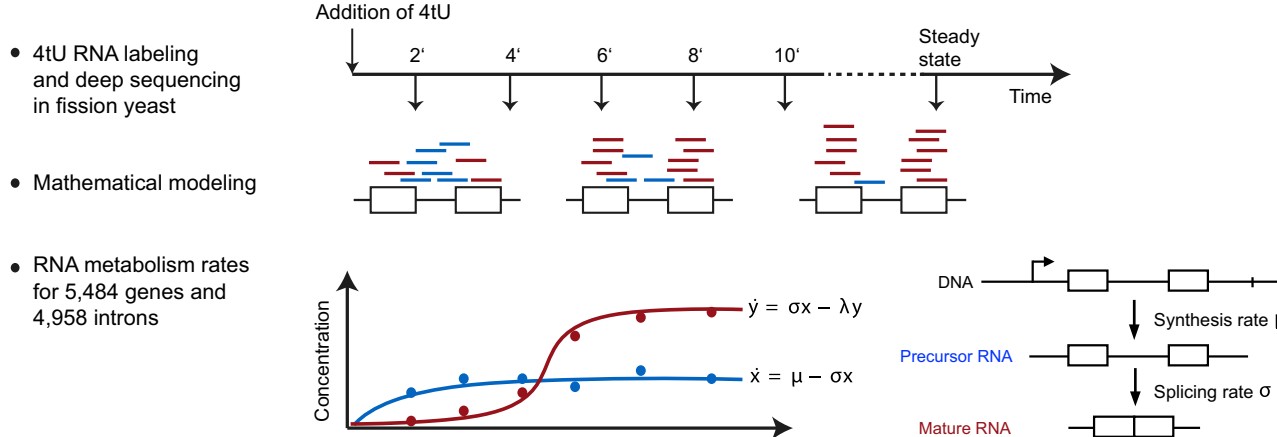

## 2. Predict RNA metabolism rates from DNA sequence

- De novo identification of predictive regulatory elements

- Quantification of single nucleotide effects on rates

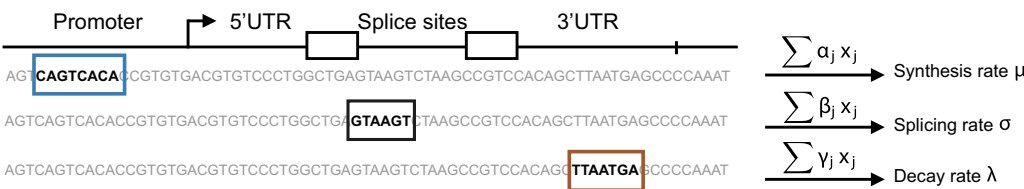

## 3. Validate regulatory elements using genetically distinct individuals

- Gene expression level across 44 strains of a recombinant strain panel (Clément-Ziza *et al* 2014)

- Matched fold-changes with predictions demonstrate causality

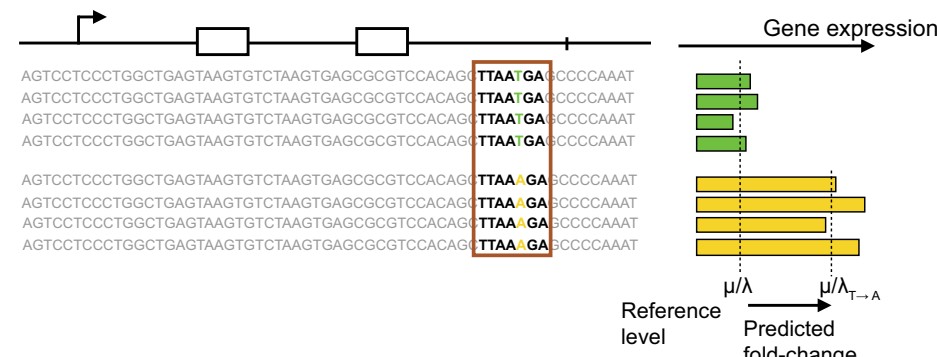

**Figure 1. Overview of the approach.**

Our approach for identifying regulatory elements that quantitatively determine RNA metabolism rates consists of three steps. In step 1 (top), genome-wide estimates of *in vivo* synthesis, splicing, and degradation rates are obtained from the analysis of 4tU RNA labeling time series. In step 2 (middle), sequence motifs (colored boxes) that are predictive for each rate are identified. The method provides for each motif and each nucleotide in a motif an estimate of its quantitative contribution to the rate. In step 3 (bottom), the elements identified in step 2, which might be predictive by mere correlation, are tested for causality. To this end, ratio of average expression levels in a population harboring the reference allele versus a population harboring a single-nucleotide variant are compared to model-predicted fold change.

frequently used 4-thiouridine, because *S. pombe* incorporates 4tU without the need of an additional transporter. In cells, the nucleobase 4tU gets efficiently converted to thiolated UTP and incorporated during transcription into newly synthesized RNAs, which can then be isolated and sequenced. To cover the typical range of synthesis, splicing, and degradation rates, cells in a steady-state culture

were harvested after 2, 4, 6, 8, and 10 min following 4tU addition. Moreover, a matching total RNA-seq was performed after 10 min labeling to control for the slower doubling time in the presence of 4tU (285 min versus 180 min). The data contained many reads that stemmed from intronic sequences and reads comprising exon–intron junctions, showing that 4tU-Seq captured short-lived precursor RNA

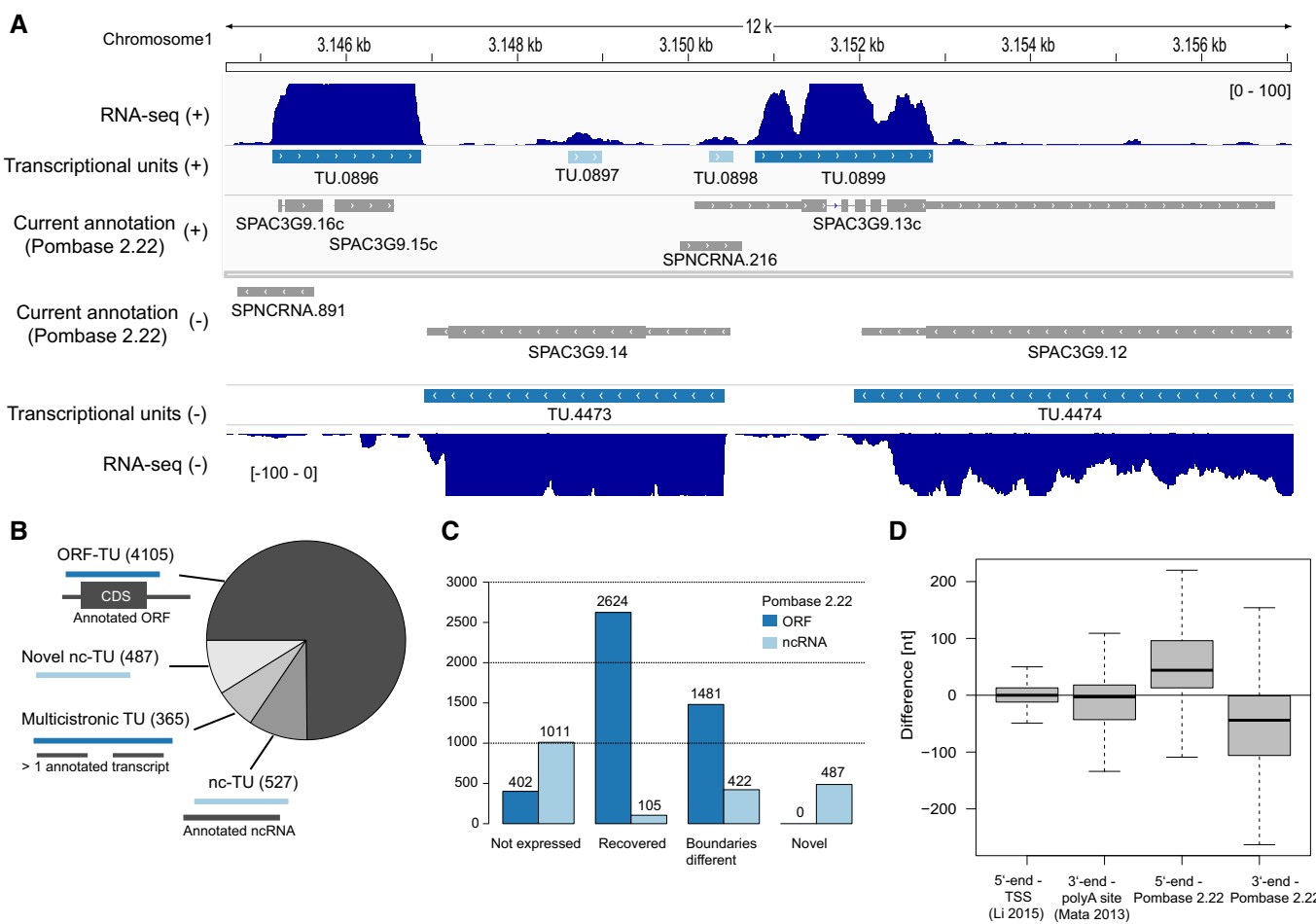

**Figure 2.    Improved annotation of transcribed loci.**

A    Example of transcribed loci annotation on a 12 kb region of chromosome 1. Data are displayed symmetrically in horizontal tracks for the plus strand (upper half) and the minus strand (lower half). From most external to most central tracks: RNA-seq per-base read coverage (marine blue), identified ORF-TUs (dark blue), identified ncTUs (light blue), and currently annotated ORFs and ncRNAs (gray, Pombase v2.22). On the current annotation tracks, UTRs are marked as thin rectangles and introns as lines. Typical changes that this study provides to the current annotation include the merging of adjacent transcripts into a single TU (e.g., SPAC3G9.16c and SPAC3G9.15c into TU.0896), the identification of novel ncRNAs (e.g., TU.0897), not recovered ncRNAs (SPNCRNA.891), and correction of aberrantly long UTRs (e.g., 3′ UTR of SPAC3G9.13c as TU.0899).

B    Classification (Materials and Methods) of the 5,484 TUs into ORF-containing (ORF-TUs), nc-TUs overlapping 70% of an annotated ncRNA (nc-TU), TUs overlapping more than one annotated TU (multicistronic TUs), and novel non-coding (Novel nc-TU).

C    From left to right: a number of currently annotated transcripts that could not be recovered are fully recovered, differ by more than 200 nt, and novel TUs for ORFs (dark blue) and ncRNAs (light blue).

D    From left to right: Quartiles (boxes) and 1.5 times the interquartile range (whiskers) of the differences between 5′-ends of ORF-TUs and transcription start site mapped by CAGE data (Li *et al*, 2015), between 3′-ends of ORF-TUs and polyA sites mapped by Mata (2013), between 5′-ends of ORF-TUs and the corresponding currently annotated 5′ UTR end, and between 3′-ends of ORF-TUs and the corresponding currently annotated 3′ UTR end.

transcripts. These reads from unspliced RNA gradually ceased during the time course (Fig 3A and B), indicating that the kinetics of RNA splicing may be inferred from the data.

To globally estimate rates of RNA synthesis, splicing, and degradation, we used a first-order kinetic model with constant rates that describes the amount of labeled RNA as a function of time (Fig 3C). We modeled splicing of individual introns, where splicing refers to the overall process of removing the intron and joining the two flanking exons. The model was fit to every splice junction using the counts of spliced and unspliced junction reads (Fig 3C and D). We included in the model scaling factors that account for variations in sequencing depth, an overall increase of

the labeled RNA fraction, cross-contamination of unlabeled RNA, and 4tU label incorporation efficiency (Materials and Methods). The model was fitted using maximum likelihood and assuming negative binomial distribution to cope with overdispersion of read counts (Robinson *et al* 2010; Anders and Huber 2010). Label incorporation efficiency by this procedure was estimated to be of 1%, consistent with independent measurements obtained by high-performance liquid chromatography (HPLC, Materials and Methods, Appendix Fig S2).

Our method yields absolute splicing and degradation rates, but provides synthesis rates up to a scaling factor common to all TUs. Absolute synthesis rates were obtained by scaling all values so that

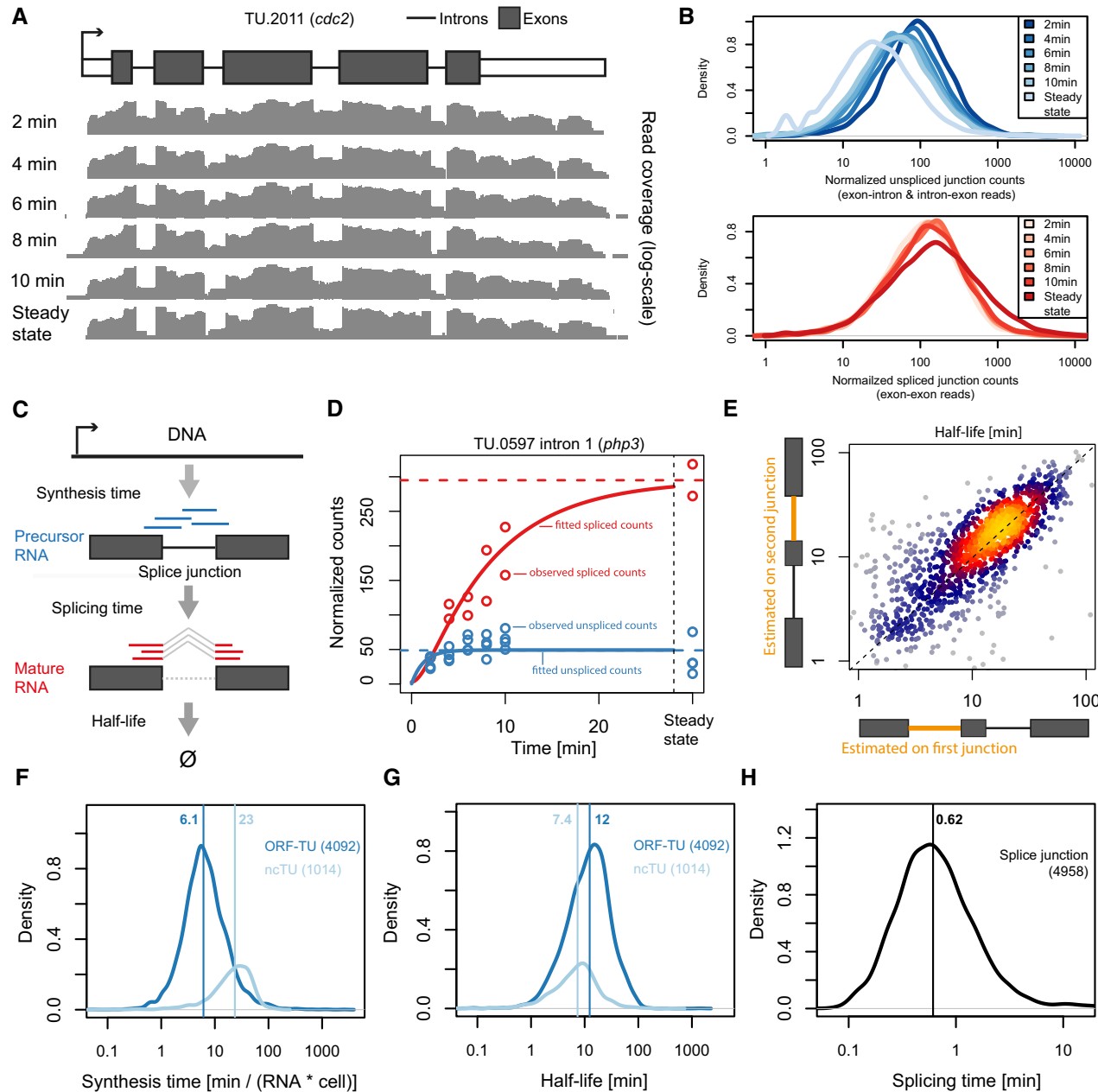

**Figure 3.  Estimating RNA processing rates using labeled RNA time series.**

A   Per-base coverage (gray tracks) in a logarithmic scale of 4tU-Seq samples at 2, 4, 6, 8, and 10 min. labeling and for one RNA-seq sample (i.e., steady state) along the UTRs (white boxes), the exons (dark boxes), and the introns (lines) of the TU encoding *cdc2*.

B   Distribution of sequencing depth normalized unspliced junction read counts (top panel) and normalized spliced junction read counts (lower panel) for the complete 4tU-Seq time series and the steady-state RNA-seq samples.

C   Schema of the junction first-order kinetics model. Each splice junction is modeled individually, assuming constant synthesis time, splicing time and half-life. Unspliced junction reads (blue) are specific to the precursor RNA and spliced junction reads (red) are specific to the mature RNA.

D   Observed (circles) and fitted (lines) splice junction counts for the first intron of TU.0597 (*php3*). Unspliced (blue) and spliced (red) normalized counts (*y*-axis) are shown for all 4tU-Seq samples and the steady-state sample (*x*-axis).

E   Half-life estimated from the first (*x*-axis) versus the second (*y*-axis) splice junction on TUs with two or more introns.

F—H   Distribution of synthesis times (F), half-lives (G), and splicing times (H) for ORF-TUs (blue) and ncTUs (light blue). Median indicated as vertical line.

the median steady-state level of ORF-TUs matches the known median of 2.4 mRNAs per cell (Marguerat *et al*, 2012). To facilitate comparisons of the obtained RNA metabolic rates, we present the synthesis rate as the average time elapsed between the production

of two transcripts in a single cell ("synthesis time"), the degradation rate as the time needed to degrade half of the mature RNAs ("half-life"), and the splicing rate as the time to process half of the precursor RNA junction ("splicing time") (Dataset EV1 and EV2).

The synthesis times and half-lives inferred from distinct splice junctions of the same TU agreed well, demonstrating the robustness of our approach (Spearman rank correlation = 0.44 for synthesis time, $P < 2 \times 10^{-16}$ and Spearman rank correlation = 0.79 for half-life, $P < 2 \times 10^{-16}$, Fig 3E and Appendix Fig S3A). Based on this comparison, we estimated the accuracy to be typically 46% for synthesis times and 31% for half-lives (mean coefficient of variation) and found that these accuracies were matched already using the two first time points and the steady-state data only (Appendix Fig S3B). Estimation of the accuracy based on comparing the estimates obtained from the two time series replicates indicate that the accuracy of the estimates of splicing times is between the accuracy for half-lives and synthesis times. The variations in the rate estimates were much smaller than the dynamic range of the rates (about 50-fold each, see below), allowing us to interpret rate differences. Supported by the good agreement of rates across junctions, we took the mean synthesis times and half-lives as estimates for the entire TU.

In order to estimate synthesis and degradation rates of intronless genes, a kinetic model that takes as input all reads overlapping the exon was used (exon model, Appendix Fig S3C). When applied to intron-containing genes, parameter estimates with the exon model were consistent with those obtained with the splice junction model (Appendix Fig S3D and E), yet less accurate as indicated by comparing rates estimated from first and second exons (Appendix Fig S3F and G). Overall, synthesis and degradation rates correlated well with previous estimates from microarray data (Sun *et al*, 2012; Spearman rank correlation = 0.45, $P < 2 \times 10^{-16}$ for synthesis rate and Spearman rank correlation = 0.74, $P < 2 \times 10^{-16}$ for half-life, Appendix Fig S3H and I), strongly supporting our rate estimation procedure.

### Distinct kinetics of mRNA and ncRNA metabolism

Overall, RNA synthesis and degradation occurred on similar time-scales (median synthesis time of 7.4 min compared to a median half-life of 11 min) and about an order of magnitude slower than splicing (median splicing time 37 s, Fig 3F–H). These results are consistent with splicing of beta-globin introns within 20–30 s as measured by *in vivo* single RNA imaging (Martin *et al*, 2013) and argue against earlier slower estimates for splicing times of 5–10 min (Singh & Padgett, 2009). Notably, ncTUs were synthesized at a significantly lower rate than ORF-TUs (median synthesis times of 23 min and 6.1 min, respectively, $P < 2 \times 10^{-16}$, Wilcoxon test) and were degraded slightly faster (median half-life of 12 min for ORF-TUs versus 7.4 min for ncTUs, $P < 2 \times 10^{-16}$, Wilcoxon test). Thus, the differences in steady-state levels of mRNAs and ncRNAs are achieved both by longer synthesis times and shorter half-lives for ncRNAs, although the differences in synthesis times dominate. Moreover, splicing time did differ significantly between the two transcript classes (median splicing time of 0.7 min for ORF-TUs versus 1.5 min for ncTUs, $P = 1.3 \times 10^{-4}$, Wilcoxon test). Transcription is known to be the major determinant of gene expression. However, among genes expressed above background level as investigated here, the dynamic ranges across the bulk of all TUs (95% equi-tailed interval) showed similar amplitudes for all three rates (53-fold for synthesis, 47-fold for half-life, and 33-fold for splicing time, Fig 3F–H). Hence, there are large and comparable variations

between genes at the level of RNA synthesis, degradation, and splicing. In the following, we first analyze the determinants for RNA synthesis and degradation and then discuss the determinants for splicing rates.

### Sequence motifs associated with RNA metabolism

We systematically searched for motifs in ORF-TU sequences that could influence RNA synthesis, splicing, and degradation rates (Fig 1, step 2). First, 6-mer motifs were identified, whose frequency in a given gene region (promoter, 5′ UTR, coding sequence, intron, 3′ UTR) significantly correlated with either rate while controlling for other 6-mer occurrences (multivariate linear mixed model, Materials and Methods). Next, overlapping motifs associating with the same rate in the same direction were iteratively merged and extended to include further nucleotides that significantly associated with the rate (Materials and Methods). We found 12 motifs that significantly associated with RNA metabolism kinetics (Fig 4A). Motifs found within TUs were strand-specific, consistent with their function as part of RNA, whereas motifs found in the promoter region (except one, CAACCA), occurred in both orientations, suggesting that they function in double-stranded DNA. These observations strongly supported the functional relevance of the discovered motifs. The number of ORF-TUs per motif ranged from 58 (ACCCTACCCT) to 765 (TATTTAT) with motifs in the 3′ UTR being the most abundant (Fig 4B).

### Determinants of high expression

Motifs that were predictive of RNA synthesis times were only found in the promoter region, further validating our approach (Fig 4A). We identified *de novo* the Homol D-box (CAGTCACA), a fission yeast core promoter element, and the Homol E-box (ACCCTACCCT), providing positive controls. In agreement with literature (Witt *et al*, 1995; Tanay *et al*, 2005), the Homol D-box and the Homol E-box motifs were enriched in ribosomal protein genes (32% and 41% of all ORF-TUs with these motifs), frequently co-occurred in promoters (Fig 4B, Fisher test, false discovery rate < 0.1) and showed strong localization preference at a distance of around 45 bp (Homol D-box) and 65 bp (Homol E-box) upstream of the TU 5′end (Fig 4C).

The 3′ UTRs of ORF-TUs with a Homol E-box were significantly depleted for all three motifs that we found to be associated with mRNA instability (FDR < 0.1, Fig 4B), indicating that the high levels of expression of these genes are achieved by a combination of efficient promoter activity and RNA-stabilizing 3′ UTRs. Both motifs associated with decreased synthesis time by 28% (Homol D-box) and 32% (Homol E-box) per motif instance (Linear regression, Figs 4F and EV1B), but also with increased half-life (50% and 31%) of the corresponding RNAs (Fig EV1A and C), likely because those RNAs are both highly synthesized and stable.

### Determinants of RNA half-life

Motifs that were predictive of RNA half-lives were found in the promoter and in UTRs. A novel AC-rich promoter motif (CCAACA) is located near the TU 5′end (Fig 4C), and associated with a decrease in half-life by 30% per motif instance (linear regression,

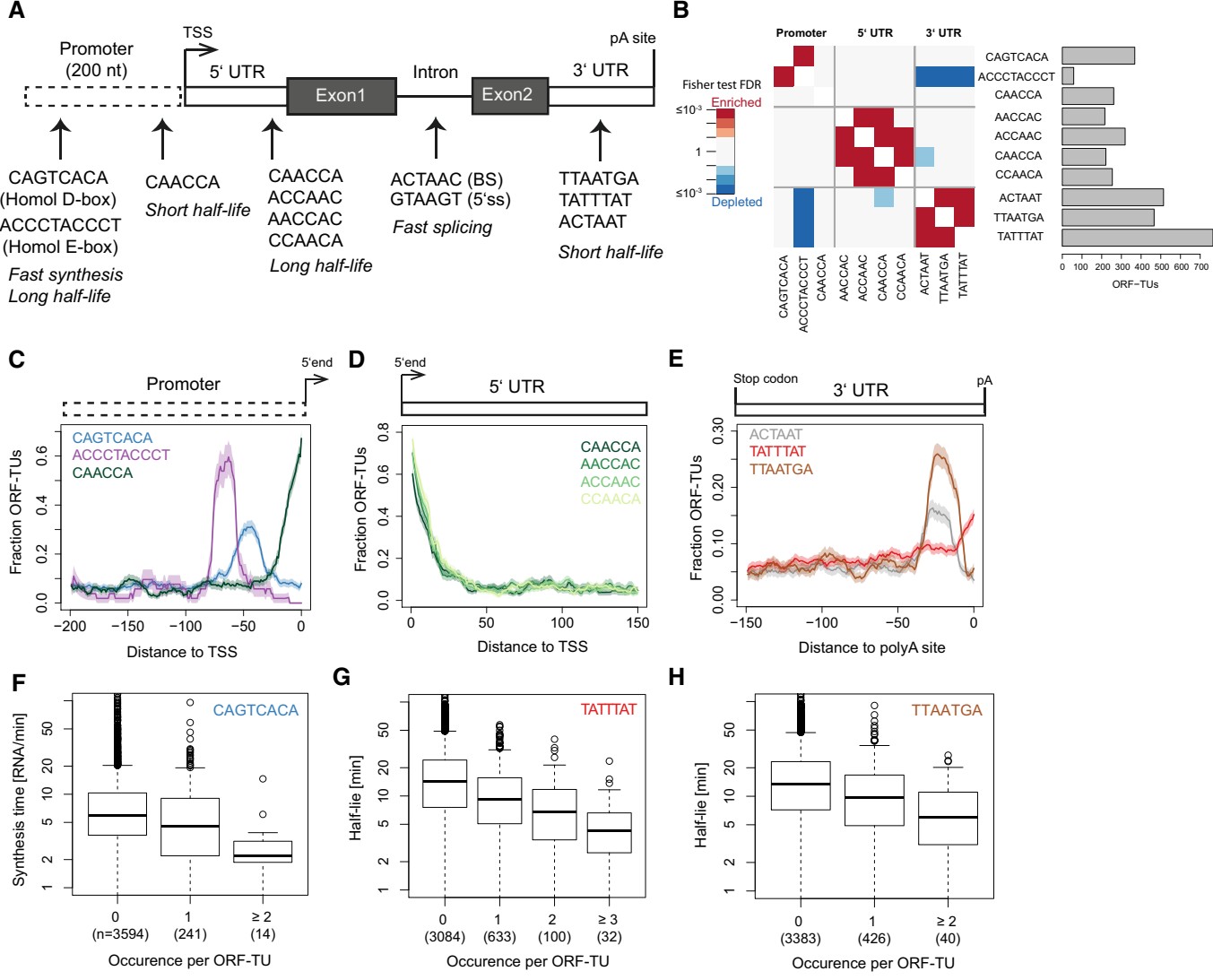

**Figure 4. Sequence motifs associated with *in vivo* degradation and synthesis rates.**

A  The 12 motifs found in promoter, 5′ UTR, intron, and 3′ UTR sequences of ORF-TUs are shown, together with their qualitative effects on RNA metabolism rates. No motif was found in coding sequences.

B  Number of ORF-TUs with at least one occurrence (horizontal bar) and significant (FDR < 0.1) co-occurrence enrichment (red) and depletion (blue) for all motif pairs. Significance was assessed using Fisher test within ORF-TUs with a mapped polyA site (Mata, 2013), followed by Benjamini–Hochberg multiple testing correction. All motif instances are provided in Dataset EV3.

C  Fraction of ORF-TUs containing the motif (y-axis) within a 20-bp window centered at a position (x-axis) upstream of the TSS for the Homol D-box (blue), the Homol E-box (purple), and the CAACCA motif (dark green).

D, E  Same as (C) for the 5′ UTR motifs (D) and for the 3′ UTR motifs with respect to polyA site (E). No positional preference was found when aligning 3′ UTRs with respect to stop codon and 5′ UTRs with respect to start codon.

F  Quartiles (boxes) and 1.5 times the interquartile range (whiskers) of the distributions of synthesis time among ORF-TUs that have zero, one or more than one occurrence of the motif CAGTCACA in their promoter sequence.

G  Quartiles (boxes) and 1.5 times the interquartile range (whiskers) of the distributions of half-lives of ORF-TUs that have zero, one, two, or more than two occurrence(s) of the motif TATTTAT in their 3′ UTR sequence.

H  Quartiles (boxes) and 1.5 times the interquartile range (whiskers) of the distributions of half-lives of ORF-TUs that have zero, one, or more than one occurrence of the motif TTAATGA in their 3′ UTR sequence.

Fig EV1D). Four AC-rich motifs were found (CAACCA, AACCAC, ACCAAC, and CCAACA) in 5′ UTRs, preferentially located near the TU 5′end (Fig 4D) and were associated with an increased RNA half-life (Fig EV1E–H). Thus, for the AC-rich motif CCAACA, the associated effect with half-life is the opposite, depending on whether the motif is located upstream or downstream of the TU 5′end.

Three motifs were detected in 3′ UTRs of ORF-TUs that all were associated with decreased RNA half-lives. One of these (TATTTAT) corresponds to the known AU-rich element (ARE) that destabilizes RNAs (Shaw & Kamen, 1986; Barreau *et al*, 2005; Cuthbertson *et al*, 2007) and that was found in 19% of the ORF-TUs and for which we estimated a half-life decrease per motif instance of 33% (Fig 4G).

The second motif (TTAATGA) and the third motif (ACTAAT) are novel and associated with a reduction in transcript half-lives by similar extents (30% and 27%, Figs 4H and EV1I). These two motifs were found in a large number of ORF-TUs (466 and 514, 11% and 13%, respectively, Fig 4B), and were co-occurring (FDR < 0.1, Fig 4B), yet not overlapping with each other. These findings suggest that TTAATGA and ACTAAT are widespread RNA elements that determine important RNA stability regulatory pathways. In contrast to the AU-rich element, the two novel 3′ UTR motifs were sharply peaking 28 bp (ACTAAT) and 25 bp (TTAATGA) upstream of the polyA site (Fig 4E), indicating that they could implicate similar mechanisms, that are distinct from the AU-rich element pathway, and that are related to RNA polyadenylation or involve interactions with the polyA tail. Two of our motifs, the AC-rich element in the promoter region, and the ACTAAT in 3′ UTRs are enriched in the same regions of human, mouse, rat, and dog genes (Xie *et al*, 2005), indicating that their function is conserved from *S. pombe* to mammals.

### Effects of single nucleotides on RNA kinetics

We next asked whether deviations from the consensus sequence of the discovered motifs can predict changes in synthesis time and half-life. We considered a linear model that included the effect of changes at each base position and the number of motifs present in each gene or RNA and fitted across all genes allowing for mismatches (Materials and Methods). Generally, deviations from the consensus sequence associate with decreased effects of the motif on synthesis time or half-life. These changes often neutralize the effect of the motif. For instance, loss of the consensus Homol D-box apparently increased synthesis time twofold (Fig 5A, purple line). A single-nucleotide deviation from the consensus Homol D-box motif by a C at the 6th position associated with a 1.6-fold increased synthesis time (Fig 5A). Similarly, a T-to-G substitution at the 5th position of the TTAATGA motif was predicted to lead to a 1.4-fold increased half-life, similar to the loss of the complete consensus motif (Fig 5B). Changes in positions flanking the motif have minor effects but may play functional roles (Fig 5A and B). Nucleotides associated with important effects tended to also be more frequent (Sequence logo, Fig 5A and B) indicating that there is evolutionary pressure on these positions and further indicating that these motifs are functional. Similar results were obtained for all motifs (Fig EV2).

### New regulatory motifs predict effects of *cis*-regulatory variants

To further provide evidence for the functional role of these new motifs, we asked whether genetic variants affecting these sequence elements resulted in a perturbed expression level in a direction and extent that match the predictions (Fig 1, step 3). We analyzed the expression data of an independent study that profiled steady-state RNA levels of a library of 44 different recombinant strains obtained from a cross between the standard laboratory strain 968, also profiled here, and a South African isolate Y0036 (Clément-Ziza *et al*, 2014). In recombinant panels, the alleles of a reference and of an alternate parental strain are randomly shuffled by meiotic recombination within the population. For a variant of interest, recombinant strains group in two subpopulations: About one half carries the reference allele and the other half the alternate allele. Variants that

are not in linkage with the one of interest, for lying on another chromosome or far away on the same chromosome, are approximately equally inherited within the two subpopulations. Hence, differential gene expression between the two subpopulations reflects local regulatory variants, such as promoter and RNA motifs, while controlling for distant, trans-acting regulatory variants.

To evaluate the effects due to perturbations of the motifs, we restricted the analysis to ORF-TUs with a variant that we predicted to significantly affect the rate (Materials and Methods), and harboring no further variant within the promoter region and the whole TU. These variants affected 20 motifs and were all single-nucleotide variants (Dataset EV5). A positive control was provided by the alternate allele of the gene *rctf1*, which differed from the reference allele by a single nucleotide, a G-to-T substitution at the third position of a Homol D-box motif in its promoter. Recombinant strains harboring the alternate allele showed significantly lower steady-state expression levels (Fig 5C, $P = 2 \times 10^{-10}$, one-sided Wilcoxon test) consistent with the predicted 1.35-fold increased synthesis time (Fig 5A).

Two variants acting in an opposite fashion strongly supported the functional role of the 3′ UTR motif TTAATGA. The linear model predicted a 1.23-fold increased half-life for a A-to-G substitution at the 7th position (7.A > G, Fig 5B). Consistently, 7.A > G substitution occurring on the gene *SPCC794.06* led to a significantly increased expression level (Fig 5D, $P = 2 \times 10^{-4}$) whereas the (7.G > A) in the gene *mug65* led to a significantly decreased expression level (Fig 5E, $P = 10^{-4}$). Among the novel motifs, the TTAATGA could be validated (3 out of 4 genes with a significant change in expression in the predicted direction $P < 0.05$) as well as the AACCAC motif (2 out of 2 genes with a significant change in expression in the predicted direction $P < 0.05$). The other motifs generally did not yield significant changes, possibly because the predicted and the observed effects were of small amplitude. In the 20 variants, the observed and predicted fold changes did not only agree in direction but also in amplitude (Pearson correlation, $P = 9 \times 10^{-4}$, Fig 5F), demonstrating the model predicted quantitatively the effects of single mutations and providing strong evidence for the functional role of these motifs.

### Intron sequences determining splicing kinetics

Sequence motifs predictive of splicing times were found only in introns, and here only in the donor region downstream of the 5′-splice site (5′SS) and at the branch site (BS). We complemented this set with the 3′-splice site (3′SS) and extended motifs in each direction as far as significant single-nucleotide effects were found (linear regression and cross-validation, Materials and Methods, Fig 6A). Significant effects were found up to six nucleotides downstream of the 5′SS. These bases are those pairing with the spliceosome component U6 small nuclear RNA during the first catalytic step of splicing (reviewed in Staley & Guthrie, 1998; Smith *et al*, 2008). We also found significant effects up to seven nucleotides 5′ of the branch point adenosine and one nucleotide 3′ of it, entailing all but one of the seven nucleotides pairing with the U2 small nuclear RNA (Smith *et al*, 2008). These two regions showed the strongest effects, with typically 1.1- to 1.5-fold decreased splicing time compared to consensus, showing that exact base-pairing with U6 and U2, although not required for splicing, are a determinant for its kinetics.

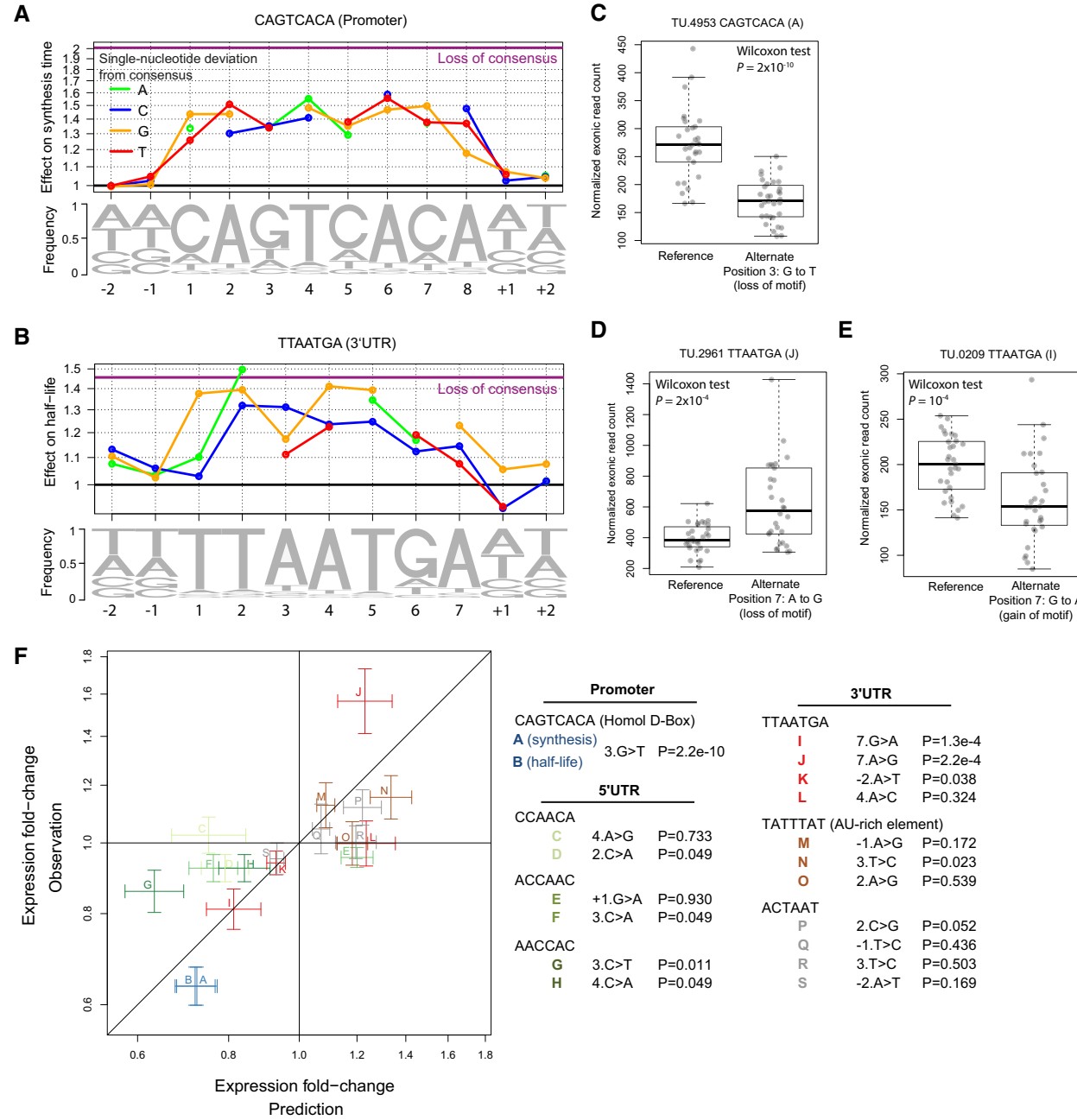

**Figure 5. Single-base substitution effects on RNA synthesis and half-life.**

A    Nucleotide frequency within motif instances (lower track) and prediction of the relative effect on synthesis time (upper track) for single-nucleotide substitution in the Homol D-box consensus motif and of complete loss of the consensus motif (purple line). Coefficients for all motifs are available in Dataset EV4.

B    As in (A) for the 3′ UTR motif TTAATGA.

C–E    Quartiles (boxes) and 1.5 times the interquartile range (whiskers) and individual data point of exonic read counts normalized for sequencing depth and batch effects (*y*-axis) for strains grouped by genotype (*x*-axis) for the gene *rctf1* (C), *SPCC794.06* (D), and *mug65* (E)

F    Validation of motifs using expression data of a recombinant strain library (Clément-Ziza *et al*, 2014). Fold change in steady-state expression level due to a single-nucleotide variant as predicted from our models (*x*-axis) against average expression fold change between strains harboring the variant and strains harboring the reference allele (*y*-axis). Estimated standard errors for the prediction (linear regression, Appendix) and the observation (Appendix) are represented by the vertical and horizontal segments. The overall Spearman rank correlation is 0.76 (*P* = 0.006). In legend: SNP code and one-sided Wilcoxon test *P*-value.

Significant but weaker effects (less than 1.1-fold) extending up to 8 nucleotides 3′ and 5′ of the 3′SS were also found.

Deviations from the consensus sequence invariably associated with increased splicing time (Fig 6A). Also, splicing time anti-correlated with the frequency of the core branch site sequence across the genome (Fig 6B). These observations indicate that there is selective pressure on all introns for rapid splicing in *S. pombe*. We then asked whether the selective strength at these positions

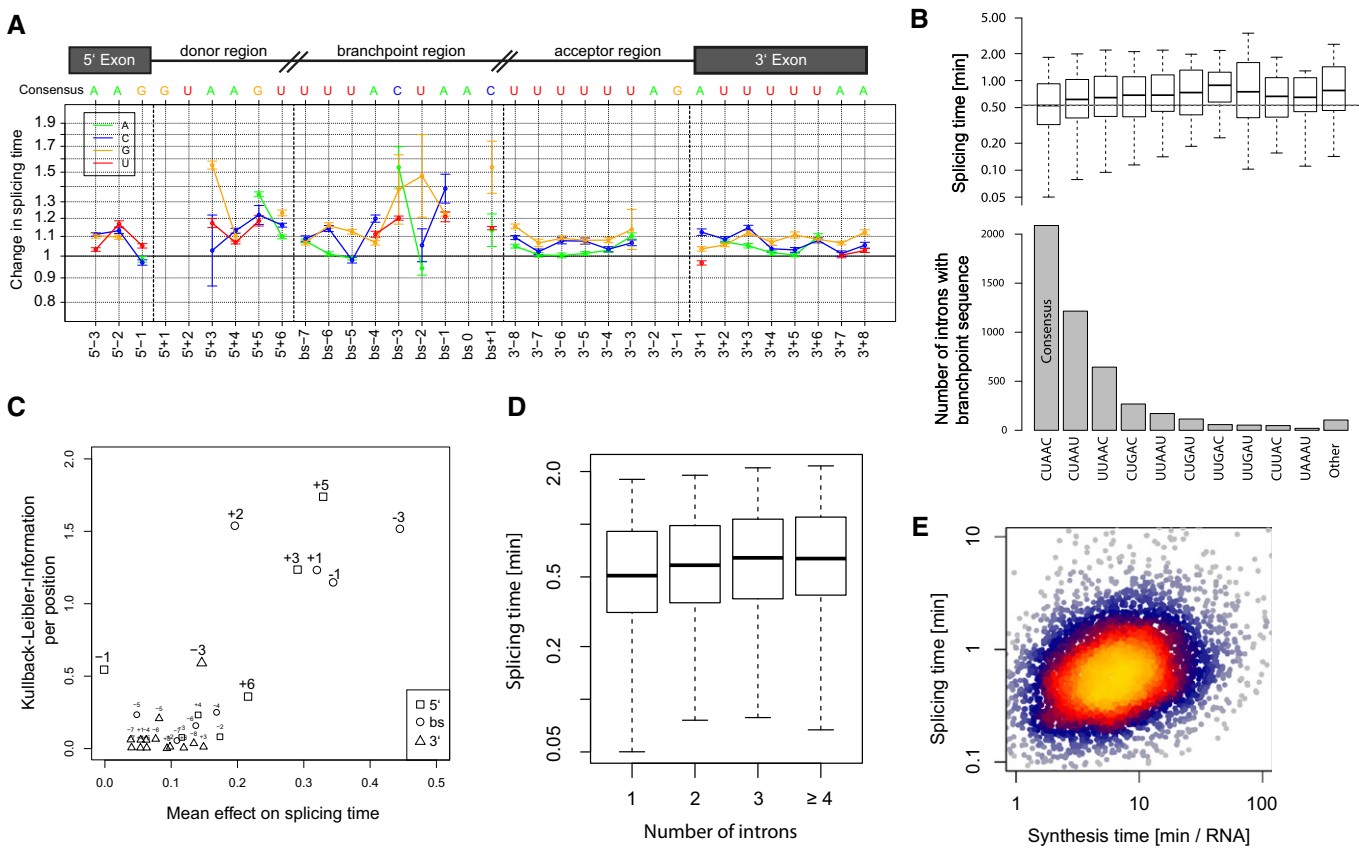

**Figure 6.  Determinants of *in vivo* splicing rates.**

A   Prediction of the relative effect and standard error (Appendix) on splicing time (*y*-axis) for single-nucleotide substitution compared to consensus sequence around the 5′splice site, the branch site, and the 3′splice site (cartoon top panel). Effects at invariant positions (5′SS: GU, BS: A and 3′SS: AG) cannot be computed. All coefficients are provided in Dataset EV4.

B   Occurrence (bottom panel) and quartiles (boxes) and 1.5 times the interquartile range (whiskers) of the distribution of half-splicing times (top panel) per BS motif (*x*-axis) sorted by frequency. The median splicing time of introns with consensus sequence is indicated with a dashed line.

C   Information content (*y*-axis) versus mean effect on splicing time (*x*-axis) for each position (relative numbers) of the 5′SS (squares), BS (circles), and 3′SS (triangles). Positions with information content > 0.3 are highlighted.

D   Distribution of splicing times (*y*-axis) versus number of introns in the TU (*x*-axis).

E   Splicing time (*y*-axis) versus synthesis time (*x*-axis)

always reflected their quantitative contribution to the rate of splicing. Overall, the mean effect of a deviation from the consensus significantly correlated with how little variable the base was across all introns genome-wide (Kullback–Leibler information, Spearman rank correlation = 0.61, $P = 5 \times 10^{-4}$, Fig 6C). Positions within the branch site region and downstream of the 5′SS are most commonly found as consensus and showed the largest effect on splicing kinetics (Fig 6C). The last nucleotide of the 5′ exon is generally a guanine but did not influence splicing time (Figs 6C, 5′SS-1 position), indicating that other sources of selection influence this position.

### Splicing kinetics also depends on RNA synthesis

Splicing time did not strongly correlate with intron length (Spearman rank correlation = 0.03, $P = 0.05$) and correlated negatively with TU length (Spearman rank correlation = $-0.16$, $P < 2 \times 10^{-16}$, Fig EV3A and B), showing that short transcripts are spliced more

slowly. This is in contrast to observations in mouse, where short transcripts and short introns are more rapidly spliced than longer ones (Rabani *et al*, 2014). This apparent discrepancy might be due to the fact that *S. pombe* neither contains very long genes nor very long introns. Splicing time increased with the number of introns (Fig 6D) as in mouse cells (Rabani *et al*, 2014), independently of the relative position of the intron within the transcript (Fig EV3C–E). However, this correlation could be explained by the fact that genes with few introns also have efficient splice site and branch site sequences (multivariate analysis and Fig EV3F). Thus, it is not the number of introns *per se* that affects splicing, rather, genes that give rise to rapidly processed RNAs evolved to have few introns and efficient splicing RNA elements.

Splicing time correlated positively with synthesis time (Spearman rank correlation = 0.28, $P < 2 \times 10^{-16}$, Fig 6E), in agreement with results in mouse (Rabani *et al*, 2014). This may be due to co-evolution of synthesis and splicing, or because highly transcribed loci are more readily accessible to the splicing machinery. This

finding is not in contradiction to the understanding that fast RNA polymerase elongation inhibits splicing (Singh & Padgett, 2009), because synthesis rate is mostly determined by the rate of transcription initiation rather than elongation (Ehrensberger *et al*, 2013). Altogether, multivariate analysis (Materials and Methods) indicated that sequence elements, synthesis time, and TU length independently enhance splicing, where sequence is the major contributor (50% of the explained variance), followed by synthesis rates (42% of the explained variance).

**Antisense transcription affects mRNA synthesis, not stability**

Repression by antisense transcription is increasingly being recognized as an important mode of regulation of gene expression, but its mechanisms remain poorly understood (Xu *et al*, 2011; Pelechano & Steinmetz, 2013). In our genome annotation, convergent TUs generally did not overlap (1,022 out of 1,616), typically leaving 75 bp of untranscribed sequence in between (Fig 7A). Among overlapping convergent pairs, TU 3′-ends were enriched within introns ($P = 0.001$) and depleted within exons ($P = 0.001$) of the opposite strand (Appendix Fig S4, 1,000 random permutations of TU pairs), likely because coding sequence is highly restrained and may impair encoding of polyadenylation and termination signals for the opposite strand.

Although transcripts are generally not antisense to each other, we found 520 ncTUs antisense of ORF-TUs (one example in Fig 7B). In fission yeast, antisense transcription could repress sense RNA synthesis, as in *S. cerevisiae* (Schulz *et al*, 2013), or affect RNA stability by RNA interference, because fission yeast, unlike budding yeast, contains the RNAi machinery. ORF-TUs with antisense ncTUs overlapping at least 40% exhibited significantly increased synthesis times (Wilcoxon test, $P = 9 \times 10^{-7}$), consistent with repression of mRNA synthesis by antisense transcription. This effect was higher when the antisense ncTU covered a larger area of the ORF-TU (Fig 7C). However, no difference regarding mRNA stability was observed (Fig 7D). Taking together, these results indicate that expression levels of those ORF-TUs were mainly regulated transcriptionally rather than by RNA interference-mediated post-transcriptional repression. The mechanism could be *cis*-acting, for example affecting the local chromatin organization (Ard *et al*, Nat. comm 2014), or *trans*-acting via the siRNA pathway (Bitton *et al* MSB 2011), which in fission yeast acts co-transcriptionally.

# Discussion

Here, by combining metabolically labeled RNA profiling at high temporal resolution with computational kinetic modeling, we

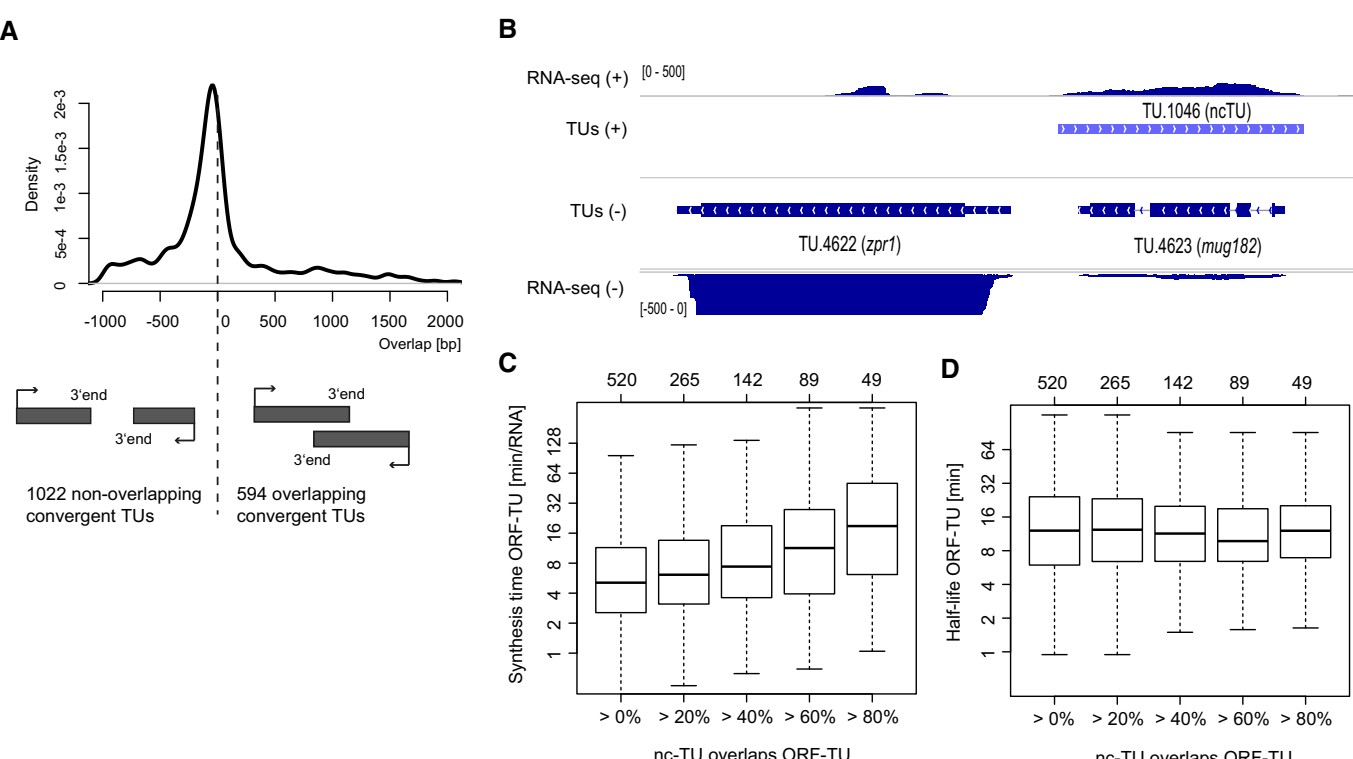

**Figure 7.  Antisense transcription represses ORF-TUs synthesis.**

A   Distribution of the overlap of the 1,616 convergent TUs separated by less than 1,000 bp. Most convergent TUs did not overlap.

B   Example of an ORF-TU (*mug182*) that is covered completely by a ncTU (TU.1046) on the opposite strand. The RNA-seq read coverage (steady-state expression) of *mug182* is considerably lower than of the adjacent gene *zpr1*.

C   Quartiles (boxes) and 1.5 times the interquartile range (whiskers) of the distribution of synthesis times of ORF-TUs grouped by the fraction of overlap by antisense ncTUs.

D   As in (C) for half-lives.

obtained *in vivo* RNA synthesis, splicing, and degradation rates across an entire eukaryotic genome, providing insights into RNA metabolism and its sequence determinants. In addition, our systematic annotation of the transcribed genome of *S. pombe* redefines most ncRNAs and a large fraction of UTRs in mRNAs, in particular 5′ UTRs and thus promoter regions. Hence, our data will be an important resource for the *S. pombe* community, and our approach will be of general use for systems biology studies of eukaryotic gene expression and its regulation.

So far, only Rabani *et al* (2014), using mouse cells, have reported a computational tool to access RNA metabolism and estimate genome-wide splicing rates. That study had used mammalian cells, resulting in a limitation in sequencing depth that restricted many parts of the analysis to the 10% most expressed splice junctions. Due to the higher sequencing depth, our analysis in *S. pombe* could be global. Another advantage of fission yeast is the absence of alternative splicing, which simplifies the analysis and makes rate estimation very robust. Multiple lines of evidence based on independent data, down to the contribution of individual base to splicing recapitulating nucleotide interactions between the precursor RNA and the spliceosome, support the high quality of our dataset.

We have estimated RNA metabolism rates using metabolic labeling. A growth time course showed that a significantly longer doubling time in 4tU by 50%. Hence, the rates we are reporting may differ from rates in standard rich media. Nonetheless, among all techniques to estimate RNA metabolism rates, 4tU is a non-severe perturbation. The alternative approaches involve transcriptional arrest, which is lethal and has strong effects on RNA metabolism due to tight coupling between synthesis and degradation (Sun *et al*, 2012).

We further introduced an approach to discover regulatory elements in the genome that combines *in vivo* quantification of RNA metabolic rates with robust regression on DNA sequence. Without using further information than simple gene architecture (promoter, UTRs, exons and introns), this approach recovered known regulatory motifs *de novo*, such as core promoter elements and the 3′ UTR AU-rich element, but also provided two novel 3′ UTR motifs, and AC-rich sequences in promoters and in 5′ UTRs. Our approach has several advantages. Conservation analysis is difficult in genomic regions that align poorly, as is often the case for regulatory regions, and give confounding results because selection can have regulatory and non-regulatory origins. By analyzing the relation between sequence and individual RNA metabolic rates, we uncouple the contribution of sequence elements to each step of RNA metabolism. Whereas standard motif enrichment analysis discriminates between two classes of data (e.g., highly versus lowly expressed), we used quantitative regression and therefore could exploit the full range of the data without applying any cutoff. Regression furthermore has the benefit to provide quantitative predictions regarding genetic perturbations that could be directly compared to expression fold changes for functional validations. Our method is general and can be applied to other organisms. Here, we analyzed steady state of a single growth condition, which might explain why we found regulatory elements such as the Homol D and the Homol E-box or the AU-rich element that are presumably constitutive. Application of the approach in a comparative fashion between different conditions could help in identifying a wider range of regulatory elements. Moreover, the approach could be extended to study other layers of gene regulation such as ribosome recruitment, or translation.

Functional evidence of the discovered motifs was obtained by exploiting existing transcription profiles of genetically distinct strains. To this end, the analysis was restricted to genes harboring a single variant across the promoter and the whole gene body. Although one cannot exclude on every single gene that further independent mutations in linkage are causative of the observed expression changes, the agreement for each motif over multiple genes in direction and amplitude strongly indicates the functionality of the motifs. Transcription profiles across genetically distinct individuals are increasingly available and include recombinant panels of model organisms such as *S. cerivisiae*, fly, mouse, *A. thaliana*, and human. Hence, our approach could help interpret the transcription profiling in human individuals. In the future, the application of our model may help to understand the consequences of regulatory variation in the human genome, with important implications for understanding gene regulation and interpreting the many disease-risk variants that fall outside of protein-coding regions (Montgomery & Dermitzakis, 2011).

# Materials and Methods

### Strains

All experiments were done with the strain ED666 (BIONEER) (h+, *ade6*-M210, *ura4*-D18, *leu1*-32).

### 4tU labeling and RNA extraction

A fresh plate (YES) was inoculated from glycerol stock. An overnight culture was inoculated (YES medium: 0.5% w/v yeast extract (Difco), 225 mg/l each of adenine, histidine, leucine, uracil, and lysine hydrochloride, 3.0% glucose) from a single colony and grown at 30°C. In the morning, a 120 ml culture (YEA medium: 0.5% w/v yeast extract (Difco), 75 mg/l adenine, 3.0% glucose) was started at $OD_{600}$ 0.1 and grown to $OD_{600}$ of 0.8 at 32°C in a water bath at 150 rpm. 4-thiouracil was added to 110 ml of culture at 5 mM final concentration. About 20 ml of samples was taken out after 2, 4, 6, 8, and 10 min. Each sample was centrifuged immediately at 32°C, at 3,500 rpm for 1 min. The supernatant was discarded, and the pellet was frozen in liquid nitrogen. All experiments were performed in two independent biological replicates. Total RNA was extracted, and samples were DNase digested with Turbo DNase (Ambion). Labeled RNA was purified as published (Sun *et al*, 2012).

### Sequencing

rRNA was depleted using the Ribo-Zero™ Gold Kit (Yeast, Epicentre) according to the manufacturer's recommendation with 1.5 μg labeled RNA and with 2.5 μg total RNA as input. Sequencing libraries for the time series samples were prepared according to the manufacturer's recommendations using the ScriptSeq™ v2 RNA-seq Library Preparation Kit (Epicentre). Libraries were sequenced on Genome Analyzer IIx (Illumina). Raw files can be downloaded from Array Express accession E-MTAB-3653. Annotation and basic statistics of the libraries are given in Dataset EV6.

### RNA-seq read mapping

Single- and paired-end RNA-seq reads were mapped to the reference genome (ASM294v2.26) with GSNAP (Wu & Nacu, 2010), allowing for novel splice site identification (Appendix).

### Transcriptional units and their classification

Transcriptional units (TUs) were identified using a min-length max-gap algorithm on binarized RNA-seq coverage track (Appendix) resulting in 5,596 TUs. Of these, 112 partially overlapped ORFs and were discarded for further analysis. The remaining final set of 5,484 TUs were classified into four disjoint classes: i) ORF-TUs entirely contain one ORF only and not more than 70% of any annotated ncRNA, ii) nc-TUs do not contain entire ORFs, overlap at least 70% of an annotated ncRNA and not more than 70% of any other annotated ncRNA, iii) novel nc-TUs do not overlap by more than 70% any annotated ncRNA and do not overlap any ORF, and iv) multi-cistronic TUs contain multiple ORFs entirely or overlap 70% of two or more annotated transcripts.

### Read counts per exon, intron and splice junctions

Counts of reads aligning completely within exons or introns were obtained with the software HTseq-counts (Anders *et al*, 2014) with settings *–stranded = yes* and *-m intersection-strict*. To count reads that map to splice sites we used HTseq with one different parameter (*-m union*) to allow counting of reads that spanned the junctions. For each intron, we defined the 5′SS as the 2-nt region that contains the last position of the upstream exon and the first position of the intron. Accordingly, we defined the 3′SS as the 2-nt region that contains the last position of the intron and the first position of the downstream exon. To distinguish spliced and unspliced junction mapping reads, a custom python script checked the cigar string of each alignment for occurrences of skipped reference bases ("N"). Alignments containing "N" and overlapped with a splice site were counted as spliced junction reads.

### Rate estimation, identification of sequence elements predictive for rates and linear regression, and analysis of recombinant strain panel

Detailed descriptions are found in the Appendix.

### Estimation of 4tU incorporation by HPLC

Cells were grown for 60 min in 4tU media so that most of the RNAs prior to 4tU exposure were degraded. For the RNA extraction, 10 μl RNA in Tris/EDTA buffer was digested for 18 h at 37°C with 1 μl Riboshredder (Epicentre) plus 1 μl bacterial alkaline phosphatase (Invitrogen). RNA was precipitated at −80°C for 5 min with 3 M sodium acetate and ice-cold 100% ethanol. After centrifugation at 13,000 rpm for 5 min at RT, the supernatant was transferred to a new tube and an additional 100 μl of ice-cold 100% ethanol was added. After 5 min at −80°C and centrifugation at 13,000 rpm for 5 min at RT, the supernatant was transferred to a new tube and the sample was dried with a SpeedVac at 45°C for about 25 min. For the HPLC, samples were run on a Kinetex 5 μm C18 100 250 × 4.6 mm

(Phenomenex) column with buffer A acetonitrile and buffer B 0.05 M triethylammonium bicarbonate, pH 8.0 in a gradient 1% A/ 99% B to 15% A / 85% B in 20 min at 1.2 ml/min. The retention times of ribouridine (rU) and 4-thiouridine (4sU) were calculated from running standards (cells metabolize 4tU into 4sU prior to incorporation into RNA). Percent of incorporation was calculated by taking the value for the area under the curve for the rU and the 4sU peaks with the software Chromeleon Chromatography Management System, version 6.80 (Thermo Scientific) and multiplying them with the extinction coefficients for 260 nm for rU or for 330 nm for 4sU and then dividing the resulting value for 4sU by the value for rU.

**Expanded View** for this article is available online.

### Acknowledgements

We are thankful to Mario Halic for helpful discussions on genome annotation and to Mathieu Clément-Ziza for data sharing and analysis advices. We thank Carlo Bäjen and Jürgen Bienert (Facility for synthetic chemistry, Max Planck Institute for Biophysical Chemistry) for help with HPLC sample preparation and calculations. PE was supported by the Deutsches Konsortium für Translationale Krebsforschung DKTK. PC was supported by the Deutsche Forschungsgemeinschaft, the Advanced Grant TRANSIT of the European Research Council, and the Volkswagen Foundation. JG was supported by the Bavarian Research Center for Molecular Biosystems and by the Bundesministerium für Bildung und Forschung through the Juniorverbund in der Systemmedizin "mitOmics" (FKZ 01ZX1405A). MB was supported by CNPq, Conselho Nacional de Desenvolvimento Científico e Tecnológico – Brasil. CD was supported by a DFG Fellowship through the Graduate School of Quantitative Biosciences Munich (QBM).

### Author contributions

PC and JG designed and supervised the research. PC conceived and designed the experiments. KCM performed the experiments. PE, LW, MB, JG, CD, and SI analyzed the data. PE, JG, PC, LW, and MB wrote the manuscript.

### Conflict of interest

The authors declare that they have no conflict of interest.

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
