## [Review Process File · Molecular Systems Biology]

Determinants of RNA metabolism in the *Schizosaccharomyces pombe* genome

Philipp Eser, Leonhard Wachutka, Kerstin C. Maier, Carina Demel, Mariana Boroni, Srignanakshi Iyer, Patrick Cramer and Julien Gagneur

Corresponding author: Julien Gagneur, Ludwig-Maximilians-Universität München

Review timeline:

Submission date:	24 August 2015
Editorial Decision:	29 September 2015
Revision received:	19 December 2015
Accepted:	10 January 2016

Editor: Thomas Lemberger

Transaction Report:

1st Editorial Decision

29 September 2015

Thank you again for submitting your work to Molecular Systems Biology. We have now heard back from the three referees who agreed to evaluate your manuscript. As you will see from the reports below, the referees find the topic of your study interesting. They raise however several points, which should be convincingly addressed in a revision of the work.

The major points raised by the reviewers refer to the following aspects:

- it appears that the probability of 4sU incorporation is a key parameter, which should be experimentally verified.
- the growth rate in presence of 4tU should be reported
- the data analysis (ie segmentation) should be described more precisely
- the datasets used in this study should be deposited in an appropriate public repository and the respective accession numbers should be explicitly included in a "Data availability" section at the end of Materials & Methods
- the functional validation of selected motifs would be a great addition but, in view of the general level of support provided by the reviewers, performing these experiments is not mandatory for publication in this case.

 REFEREE COMMENTS

Reviewer #1:

The paper presents a high-resolution time course of RNA synthesis, processing and degradation in the fission yeast *S. pombe*. This is done by profiling newly synthesized RNA, which is labeled in

vivo with thio-uracil. From these data the authors improve genome annotation and estimate synthesis, decay and splicing rates. The data are then mined to identify motifs and polymorphisms associated with synthesis and degradation kinetics.

In general this is a very good paper. The dataset produced is of very high quality, and the analysis is comprehensive and novel. The data will be very useful for the fission yeast research community, but the methodology and the analysis will also be of interest but researchers outside this field. There are a few issues with the experimental setup that require clarification. I also request two simple experiments that I consider essential.

General remarks:

There are some key issues with the experimental setup: First, the authors have chosen a strain with multiple auxotrophic markers (not ideal, as these mutations may have effects on cell growth even in the presence of the corresponding supplements). Crucially, one of the markers they use is *ura4-D18*, required for the synthesis of uracil. Second, the authors use thio-uracil (rather than thio-uridine, as has been done in their previous experiments in *S. pombe*). Third, the authors use rich medium (YES) rather than minimal medium, and thus the actual concentration of uracil in the medium is unknown. These issues raise a number of questions (see below).

The reannotation of the fission yeast genome is very useful, but some of the methods used are a bit crude. For example, the fact that two coding sequences appear contiguous on RNA-seq data (as the example shown in figure 2) does NOT mean that they are part of a single polycistronic transcription unit. Much more likely, there are two independent transcripts that partially overlap (one would need to do a Northern to be sure, but simple sequence analyses of the coding sequences suggest that they are independent).

Major point:

The authors have previously shown (Miller et al) that the probability $p(4sU)$ that one transcript incorporates at least one 4sU need to be taken into account to avoid significant biases. As far as I can see this has been estimated from the data but not calculated experimentally (extended view, 3.4). Although this value is similar to the one previously measured by the authors (Miller et al), the latter had been obtained in a different species and using a different labeled molecule. Given the importance of this value, it should be validated experimentally under the conditions used in this paper (if the experimentally-obtained parameter is similar to that estimated by the authors, it would not be necessary to reanalyze the data, simply to confirm that the estimates are reasonable).

Minor points:

- 1] I may have missed, but I could not find a reference to a database deposition for the sequencing data (ArrayExpress / GEO). This should of course be supplied
- 2] Do these cells grow normally (in periods of time of up to a generation time) in the presence of 4tU? These should be addressed by measuring cell growth in a time course under the conditions used for the labeling.
- 3] Is the *ura4-D18* mutation required for efficient labeling?
- 4] The use of rich medium is unfortunate, as batch-to-batch variations in uracil concentration might affect the labeling kinetics (note that this does not invalidate the study, but the design could have been better).
- 5] It is possible that there is a 'lag' time before the cells start to incorporate 4tU. This might be significant for very short labeling times. Is there any experimental data on this? Could the authors comment on this?
- 6] Is there any experimental reason for using thio-uracil rather than thio-uridine?
- 7] The exact composition of YES should be given, as it is a key experimental parameter.
- 8] Page 11. The *S. pombe* Zfs1 protein binds to AU-rich elements and has been extensively characterized by the Blackshear group and others. This should be cited.
- 9] A relevant paper in the discussions of anti-sense RNAs is Bitton et al. Mol Syst Biol. 2011, it should be cited.
- 10] The limitations of the reannotation of the *S. pombe* genome (such as the merging of adjacent coding sequences into a single transcriptional unit) should be acknowledged and discussed.
- 11] There is a huge body of literature on RNAi and silencing in *S. pombe*. The findings on the effect of anti-sense RNAs should be put in the context of the relevant literature.

Reviewer #2:

In this manuscript Gagneur and colleagues perform in-depth and global investigation of the *S. pombe* transcriptome. They combine computational methods with sequencing of 4-thiouridine labeled RNA from 2 to 10 minutes to determine transcription units (TUs), synthesis, splicing, and degradation rates. Using the measured kinetic parameters, they relate specific sequence motifs to individual aspects of RNA metabolism. Support for the functionality of these sequence motifs is gained by comparing changes in motif content to changes in gene expression across a panel of recombinant *S. pombe* strains. Finally, they show the proportion of overlap between ORF-TU and an antisense nc-TU correlates with the increase in time to synthesize the ORF-TU.

The manuscript is well-written and should be of general interest. This goes beyond genomics and MSB is an appropriate venue for this work. The computational aspects of the work are quite strong and generally sufficiently described. The data and analysis provides a comprehensive resource for researchers interested in gene expression mechanisms and/or *S. pombe*.

Major

1) There is not sufficient information describing the data-sets. For each data set, single/paired, what was the sample, how many reads, how many aligned...

2) The authors need to provide more specifics regarding the segmentation algorithm. The threshold above background is 10.25 in the figure S1B. The background in this case is coverage over non-annotated regions. The coverage of the annotated region shows a bimodal distribution, for which the trough is ~ 20 . What are the results if this is used as a cut-off? Especially since the authors actually perform an additional filtering step anyway at a similar coverage in 2 minute 4tU-Seq data (there is a parenthesis missing):

"To further improve this map of TUs, we only kept TUs that showed significant read coverage (average per base coverage < 20 in the two-minute labeled 4tU-Seq samples, normalized for sequencing depth using annotated ORFs read counts and following Love et al. (2014))."

3) It would be important to evaluate the number of time points necessary to glean similar biological insights. How do the derivatives at 2,4,6,8 and 10 minutes (which represent synthesis, splicing, and degradation rates) compare to one another for given gene? To this end can the authors provide a boxplot and/or correlation the distribution of these rates. Additionally, this may reveal interesting kinetic aspects of RNA metabolism.

4) The rate estimation for the intronless TUs is a bit confusing. It is unclear why/how sigma is being evaluated given there is no processing.

5) It is unclear how the branch point is being defined.

Minor

1) Please provide the % strandedness of the RNA-seq libraries generated. Related to this, to what extent do highly expressed TUs have a "shadow" on the opposite strand?

2) Do the multicistronic TUs contain two distinct non-overlapping ORFs? Please provide more description of these TUs.

"The remaining 365 TUs contained two or more annotated adjacent transcripts, and thus may be multicistronic RNAs."

3) I have difficulty with interpreting 3D plots in 2D...could the authors provide a histogram of TU lengths for the final TU list. Additionally, it would be helpful to provide histograms of intron number and lengths for these TUs, especially for researchers not familiar with *S pombe*.

4) The figure 5 appears twice.

Reviewer #3:

In this manuscript Eser and colleagues report transcription, splicing and degradation rates for the complete transcriptome of fission yeast during rapid proliferation. This rich dataset will be very useful for the fission yeast community. In addition, this study provides insight into the biology of fission yeast gene expression including differences in expression kinetics between mRNA and ncRNA, correlation of splicing and transcription rates, regulation of antisense expression by transcriptional interference. Moreover, the author developed a computational approach to infer regulatory motif functionality from RNA-seq data using a recombinant strains library. This is overall a very interesting paper. Some specific comments are listed below:

1) P.5: "TUs that did not show significant signal in the 4tU-Seq dataset were considered as artifacts and discarded...". It is unclear to me how the authors make the distinction between "artefacts" and TU that are not expressed in rich media (as discussed on page 6)? This remark is also valid for ncRNA many of which are condition specific and/or actively degraded by the RNA surveillance machinery.

2) P.6: The authors provide new boundaries for a series of transcript in fission yeast most changes leading to shorter UTRs. Calling gene structure from RNA-seq data depends heavily on the set of cutoffs that are used for segmentation. Higher cutoffs leading to shorter transcript as it is clear from figure 1A. The good concordance with the poly(A) mapping data published by Mata is a good sign. It would be useful, I think, that the authors also validate their 5' UTR calls using the CAGE dataset published by Li et al (PMID:25747261). It will also be important that the authors get in touch with the pombase team to ensure their data are incorporated as a track in the database. This will make them visible to the community. Finally, the authors' use of the term "revised annotation" is too strong. As far as I am aware, UTR coordinates in fission yeast have not been incorporated formally into the gene structures.

3) P.6: I struggle to understand how spurious antisense signal can lead to extended sense 3' UTR boundaries? The coordinates available in pombase are primarily the longest UTR calls published by Rhind et al. These were derived from strand-specific data.

4) The number of regulatory sequences identified in promoters seems rather low. Why couldn't more transcription factor binding sites be identified? Is it due to the sensitivity of the approach or does it rather suggest that sharing TF binding sites does not often lead to similar transcription rates? On a related note, does the rather subtle effects of mutations in the RNA motifs reflect technical limitations or is it evidence that regulation of RNA stability at steady-state is a fine-tuning mechanism?

5) The recombinant strain approach used to provide evidence of the new motifs functionality is clearly elegant. However, the impact of this paper would be significantly increased if the authors were to include functional validation of a selection of motifs and of their mutated alleles in the same genetic background using a gene reporter assay.

6) P.14: "...showing that short transcripts are spliced more slowly...". How do ribosomal proteins behave? They are small, highly expressed, and very often contain introns. Their mRNA expression seems to be optimised at every level. Is splicing an exception to this rule?

7) Is the increase in splicing rates with transcription rate related to the increase splicing efficiency with expression levels reported on figure 4 of Wilhelm et al (Nature, 2008)?

Editor:

The major points raised by the reviewers refer to the following aspects:

- it appears that the probability of 4sU incorporation is a key parameter, which should be experimentally verified.*
- the growth rate in presence of 4tU should be reported*
- the data analysis (ie segmentation) should be described more precisely*
- the datasets used in this study should be deposited in an appropriate public repository and the respective accession numbers should be explicitly included in a "Data availability" section at the end of Materials & Methods*
- the functional validation of selected motifs would be a great addition but, in view of the general level of support provided by the reviewers, performing these experiments is not mandatory for publication in this case.*

Answer:

We thank the reviewers for their useful inputs. We have addressed their concerns as detailed below. In particular, we have obtained independent experimental measures of the labeling incorporation by HPLC that are consistent with our estimations from the 4tU-seq data. Moreover, the manuscript now makes use of the Expanded View features. We think that the manuscript has very much improved.

Reviewer #1:

*The paper presents a high-resolution time course of RNA synthesis, processing and degradation in the fission yeast *S. pombe*. This is done by profiling newly synthesized RNA, which is labeled in vivo with thio-uracil. From these data the authors improve genome annotation and estimate synthesis, decay and splicing rates. The data are then mined to identify motifs and polymorphisms associated with synthesis and degradation kinetics.*

In general this is a very good paper. The dataset produced is of very high quality, and the analysis is comprehensive and novel. The data will be very useful for the fission yeast research community, but the methodology and the analysis will also be of interest but researchers outside this field. There are a few issues with the experimental setup that require clarification. I also request two simple experiments that I consider essential.

General remarks:

*There are some key issues with the experimental setup: First, the authors have chosen a strain with multiple auxotrophic markers (not ideal, as these mutations may have effects on cell growth even in the presence of the corresponding supplements). Crucially, one of the markers they use is *ura4-D18*, required for the synthesis of uracil. Second, the authors use thio-uracil (rather than thio-uridine, as has been done in their previous experiments in *S. pombe*). Third, the authors use rich medium (YES) rather than minimal medium, and thus the actual concentration of uracil in the medium is unknown. These issues raise a number of questions (see below).*

The reannotation of the fission yeast genome is very useful, but some of the methods used are a bit crude. For example, the fact that two coding sequences appear contiguous on RNA-seq data (as the example shown in figure 2) does NOT mean that they are part of a single polycistronic transcription unit. Much more likely, there are two independent transcripts that partially overlap (one would need to do a Northern to be sure, but simple sequence analyses of the coding sequences suggest that they are independent).

Major point:

The authors have previously shown (Miller et al) that the probability $p(4sUI)$ that one transcript incorporates at least one 4sU need to be taken into account to avoid significant biases. As far as I can see this has been estimated from the data but not calculated

experimentally (extended view, 3.4). Although this value is similar to the one previously measured by the authors (Miller et al), the latter had been obtained in a different species and using a different labeled molecule. Given the importance of this value, it should be validated experimentally under the conditions used in this paper (if the experimentally obtained parameter is similar to that estimated by the authors, it would not be necessary to reanalyze the data, simply to confirm that the estimates are reasonable).

Answer:

We have performed quantifications of proportion of labeled uracil using High Pressure Liquid Chromatography (First paragraph, page 8 and new Appendix Figure S2). Estimates across 3 biological and 2 technical replicates vary between 0.3% and 2.6%, supporting our original estimation of 1%. Moreover, the number of genes whose rate estimates effectively depend on the correction for labeling incorporation is not large: 584 TUs (10%) have less than 160 Us and thus a net probability of incorporation of at least one labeled U smaller than 0.8. Of these, the vast majority (559) are non-coding. Hence, the correction concerns essentially the rates of the short non-coding TUs. The claim that ncTUs are synthesized at a significantly lower rate than ORFTUs would also hold if no correction were applied, since the correction leads to higher synthesis rates estimates. It is also in line with our understanding of non-coding transcription.

Minor points:

1] I may have missed, but I could not find a reference to a database deposition for the sequencing data (ArrayExpress / GEO). This should of course be supplied

Answer: The raw sequencing data can be accessed under accession E-MTAB 3653 at ArrayExpress. We have included the accession number in the methods section. For review purpose, the data can be already accessed with login name “Reviewer_E-MTAB-3653” and password “jyuhypq”.

1] Do these cells grow normally (in periods of time of up to a generation time) in the presence of 4tU? These should be addressed by measuring cell growth in a time course under the conditions used for the labeling.

Answer: We performed a growth time-course as suggested and observed a doubling time about 50% longer (ca. 180 min in rich media without 4tU versus ca. 285 min. in presence of 4tU). Note that the total RNA-seq data were also collected in presence of 4tU (after 10 min. exposure). Hence, the dataset is consistent and the rates we are reporting correspond to the rates for growth in presence of 4tU. Nonetheless, among all techniques to estimate RNA metabolism rates, 4tU is a non-severe perturbation. The alternative approaches involve transcriptional arrest, which is lethal and has strong effects on RNA metabolism due to tight coupling between synthesis and degradation (Sun et al. Genome Research, 2012). These points are now in the results (page 7) and discussion section (page 17).

3] Is the ura4-D18 mutation required for efficient labeling?

Answer: We have never labeled a *S. pombe* strain without the ura mutation with this particular protocol. It may make labeling more efficient if the strain cannot synthesize uracil but we have no data on this.

4] The use of rich medium is unfortunate, as batch-to-batch variations in uracil concentration might affect the labeling kinetics (note that this does not invalidate the study, but the design could have been better).

Answer: This is indeed possible. We have performed two replicate time series. Within each time series, the cells were grown in the same flask so there is no growth media batch-to-batch variation.

5] It is possible that there is a 'lag' time before the cells start to incorporate 4tU. This might be significant for very short labeling times. Is there any experimental data on this? Could the authors comment on this?

Answer: We are not aware of experimental data on this. The lag is a constant that is the same for all genes. This time has to be very short since we can measure labeled RNA after 2 min labeling and thus we expect the effect to be small. As pointed out by this reviewer, getting into shorter durations might require considering estimations of that duration. We have added this point to modeling assumptions in the Appendix (section 3.10).

6] *Is there any experimental reason for using thio-uracil rather than thio-uridine?*

S. pombe, like *S. cerevisiae*, can incorporate thiouracil on its own. To take up thiouridine, both need an additional transporter (e.g. the human Equilibrative nucleoside transporter 1 ENT1). We added one sentence in the results section that gives that explanation (page 7). Thiouracil is also considerably cheaper than thiouridine.

7] *The exact composition of YES should be given, as it is a key experimental parameter.*

Answer: we have added this to Materials and Methods.

8] *Page 11. The S. pombe Zfs1 protein binds to AU-rich elements and has been extensively characterized by the Blackshear group and others. This should be cited.*

Answer: we now refer at page 11 to the article characterizing Zfs1 RNA binding (Cuthbertson et al., 2007).

9] *A relevant paper in the discussions of anti-sense RNAs is Bitton et al. Mol Syst Biol. 2011, it should be cited.*

We thank this reviewer for pointing to this reference and stressing the importance of discussing our result in the context of the RNAi silencing literature of fission yeast (See also point 11). Bitton et al show that for 4 select meiotic genes, antisense repression acts in trans. It does so in an Ago1-, but also Dcr1-, and Rdp1-dependent fashion. These observations are not in contradiction with ours because in *S. pombe* siRNA silencing also acts transcriptionally (review in Holoch and Moazed, Nature reviews genetics, 2015) through positive feedbacks involving the Dicer (Dcr1) and the RITS complex (Ago1). The antisense results are now discussed in this context (page 16).

10] *The limitations of the reannotation of the S. pombe genome (such as the merging of adjacent coding sequences into a single transcriptional unit) should be acknowledged and discussed.*

Answer: We added a sentence that stresses that these TUs may contain independently transcribed regions (top of page 6).

11] *There is a huge body of literature on RNAi and silencing in S. pombe. The findings on the effect of anti-sense RNAs should be put in the context of the relevant literature.*

Repression of heterochromatin genes is known to be mostly due to epigenetic repression by this mechanism (review in Holoch and Moazed, Nature reviews genetics, 2015). However, this is unlikely to be the mechanism acting on euchromatin genes because hallmarks of this pathway (sRNAs and H3K9 methylations are limited to very few chromatin genes (about 20, Zofall et al. Science 2011)). Hence, the association with low synthesis rates might be better explained by transcriptional interference as in the case of *tgpl* (Ard et al, Nat. comm 2014). This interpretation is now provided at page 16.

Reviewer #2:

In this manuscript Gagneur and colleagues perform in-depth and global investigation of the S. pombe transcriptome. They combine computational methods with sequencing of 4-thiouridine labeled of RNA from 2 to 10 minutes to determine transcription units (TUs), synthesis, splicing, and degradation rates. Using the measured kinetic parameters, they relate specific sequence motifs to individual aspects of RNA metabolism. Support for the functionality of these sequence motifs is gained by comparing changes in motif content to

changes in gene expression across a panel of recombinant *S pombe* strains. Finally, they show the proportion of overlap between ORF-TU and an antisense nc-TU correlates with the increase in time to synthesize the ORF-TU.

The manuscript is well-written and should be of general interest. This goes beyond genomics and MSB is an appropriate venue for this work. The computational aspects of the work are quite strong and generally sufficiently described. The data and analysis provides a comprehensive resource for researchers interested in gene expression mechanisms and/or *S pombe*.

Major

1) There is not sufficient information describing the data-sets. For each data set, single/paired, what was the sample, how many reads, how many aligned...

Answer: We added the missing information as a supplemental table (Dataset EV6).

2) The authors need to provide more specifics regarding the segmentation algorithm. The threshold above background is 10.25 in the figure S1B. The background in this case is coverage over non-annotated regions. The coverage of the annotated region shows a bimodal distribution, for which the trough is ~ 20. What are the results if this is used as a cut-off? Especially since the authors actually perform and additional filtering step anyway at a similar coverage in 2 minute 4tU-Seq data (there is a parenthesis missing):

"To further improve this map of TUs, we only kept TUs that showed significant read coverage (average per base coverage < 20 in the two-minute labeled 4tU-Seq samples, normalized for sequencing depth using annotated ORFs read counts and following Love et al. (2014))."

Answer: We have substantially extended the description of the method to identify the coverage cutoff in the Appendix (section 2). The filtering based on the 2 min labeled data was done to increase the reproducibility of the annotation across datasets. A coverage cutoff on one dataset only does not handle the reproducibility issue.

As suggested, we performed the segmentation and annotation based on a cut-off at 20 and compared the resulting TUs to the Pombase annotation and to the 5' and 3' annotations from Li et al. (2015) and Mata (2014). The plots are attached to this response (Figure 1 and 2). We would report slightly more changes to the Pombase annotation as we are doing now. Hence, we prefer to stay with our current approach.

Figure 1. Same as Figure 2D for a coverage cutoff at 20 only.

Figure 2. Same as Figure 2C for a coverage cutoff at 20 only.

3) *It would be important to evaluate the number of time points necessary to glean similar biological insights. How do the derivatives at 2, 4, 6, 8 and 10 minutes (which represent synthesis, splicing, and degradation rates) compare to one another for given gene? To this end can the authors provide a boxplot and/or correlation the distribution of these rates. Additionally, this may reveal interesting kinetic aspects of RNA metabolism.*

Answer: we have performed analysis using the first time point only (and steady-state), the two first time points etc. Comparison shows that there is very good agreement in the inferred rates between the different time points (see attachment). This analysis shows that rates could have been obtained with the two first time points only and the steady-state. The mean coefficient of variation as a function of the number of time points used is now provided in Appendix Figure S3B.

4) *The rate estimation for the intronless TUs is a bit confusing. It is unclear why/how sigma is being evaluated given there is no processing.*

Answer: we agree that using sigma for degradation rate of intronless genes is confusing. We are now using lambda.

5) *It is unclear how the branch point is being defined.*

Answer: the branch points were mapped using the FELINES algorithm (Drabenstot et al 2003). We have included this point in the Appendix, section 6.

Minor

1) *Please provide the % strandedness of the RNA-seq libraries generated. Related to this, to what extent do highly expressed TUs have a "shadow" on the opposite strand?*

Answer: This is now reported in the new table Dataset EV6. The % of strandedness is high, varying between 92.5 and 98.0%. Note that spurious antisense signal lead to underestimation of the repressive effect of antisense transcription on sense synthesis rates.

2) *Do the multicistronic TUs contain two distinct non-overlapping ORFs? Please provide more description of these TUs.*

"The remaining 365 TUs contained two or more annotated adjacent transcripts, and thus may be multicistronic RNAs."

Answer: Multicistronic TUs contain two or more annotated features, which can be genes with

ORFs or non-coding RNAs. The majority of those TUs contain two ORFs, with non-overlapping predicted CDS regions. Due to limitations of our data, we cannot resolve if these TUs represent a single or multiple transcript(s). We added a sentence on the limitations of our annotation approach to the result sections at the top of page 6. See also the response to the reviewer 1 on this point.

3) I have difficulty with interpreting 3D plots in 2D...could the authors provide a histogram of TU lengths for the final TU list. Additionally, it would be helpful to provide histograms of intron number and lengths for these TUs, especially for researchers not familiar with S pombe.

Answer: We added those histograms as Appendix Figure S1F-H.

4) The figure 5 appears twice.

Answer: The duplicate has been removed.

Reviewer #3:

In this manuscripts Eser and colleagues report transcription, splicing and degradation rates for the complete transcriptome of fission yeast during rapid proliferation. This rich dataset will be very useful for the fission yeast community. In addition, this study provides insight into the biology of fission yeast gene expression including differences in expression kinetics between mRNA and ncRNA, correlation of splicing and transcription rates, regulation of antisense expression by transcriptional interference. Moreover, the author developed a computational approach to infer regulatory motif functionality from RNA-seq data using a recombinant strains library. This is overall a very interesting paper. Some specific comments are listed below:

1) P.5: "TUs that did not show significant signal in the 4tU-Seq dataset were considered as artifacts and discarded...". It is unclear to me how the authors make the distinction between "artefacts" and TU that are not expressed in rich media (as discussed on page 6)? This remark is also valid for ncRNA many of which are condition specific and/or actively degraded by the RNA surveillance machinery.

Answer: We shall distinguish two kinds of artifacts: false negatives and false positives. Regarding false negatives, we agree with this reviewer that our annotation is blind to transcripts that are specifically expressed in other conditions, as we indeed mention in page 6. Further false negatives would include cryptic transcripts that are only revealed in mutants. The artifacts we are concerned by are suspected false positives. Some regions showing above background signal in the dataset used for segmentation did not show high signal in the 4tU time series. A colleague, Mario Halic, with whom we shared a preliminary annotation, also reported this issue. We thus decided to keep only TUs that were supported by the 4tU-seq series as well.

2) P.6: The authors provide new boundaries for a series of transcript in fission yeast most changes leading to shorter UTRs. Calling gene structure from RNA-seq data depends heavily on the set of cutoffs that are used for segmentation. Higher cutoffs leading to shorter transcript as it is clear from figure 1A. The good concordance with the poly(A) mapping data published by Mata is a good sign. It would be useful, I think, that the authors also validate their 5' UTR calls using the CAGE dataset published by Li et al (PMID:25747261). It will also be important that the authors get in touch with the pombase team to ensure their data are incorporated as a track in the database. This will make them visible to the community. Finally, the authors' use of the term "revised annotation" is too strong. As far as I am aware, UTR coordinates in fission yeast have not been incorporated formally into the gene structures.

Answer: We thank the reviewer for these suggestions. The CAGE study by Li et al. is indeed an important dataset to validate the 5' ends of our TUs. It shows that we are in very good agreement to TSS sites with a mean difference of 3 nucleotides. We added this comparison to Figure 2D. We have taken contact with Pombase to incorporate our annotation as a track. Indeed, the term

‘revised’ annotation was too strong and we have modified the text accordingly (page 6, “Improved annotation of transcribed regions in *S. pombe*”).

3) *P.6: I struggle to understand how spurious antisense signal can lead to extended sense 3' UTR boundaries? The coordinates available in pombase are primarily the longest UTR calls published by Rhind et al. These were derived from strand-specific data.*

Answer: Figure 2A helps visualizing the issue. SPAC3G9.13c has a surprisingly long 3'UTR, which turns out to be antisense to the gene SPAC3G9.12. Rhind et al indeed used a strand-specific protocol. However, these authors did not use actinomycin D to block spurious second-strand synthesis during reverse transcription (Perocchi et al, NAR, 2007). Hence, they had likely antisense artifacts leading to the long 3'UTR annotation of this gene.

4) *The number of regulatory sequences identified in promoters seems rather low. Why couldn't more transcription factor binding sites be identified? Is it due to the sensitivity of the approach or does it rather suggest that sharing TF binding sites does not often lead to similar transcription rates?*

Answer: We think this is due to the fact that we look at single growth condition. Further TFs would be found only when contrasting different growth conditions. We are discussing this issue now (page 18, 1st paragraph) and thank this reviewer for raising this concern.

On a related note, does the rather subtle effects of mutations in the RNA motifs reflect technical limitations or is it evidence that regulation of RNA stability at steady-state is a fine-tuning mechanism?

Answer: The issue is the effect size of these mutations, rather than the sensitivity. Indeed the effects match in amplitude the predictions made, showing that we are not suffering from sensitivity. One should also note that natural variants have typically small effects. These are single base mutations. A whole deletion would be predicted more deleterious (2-fold).

5) *The recombinant strain approach used to provide evidence of the new motifs functionality is clearly elegant. However, the impact of this paper would be significantly increased if the authors were to include functional validation of a selection of motifs and of their mutated alleles in the same genetic background using a gene reporter assay.*

Answer: We thank this reviewer for appreciating the re-use of the published eQTL data, which together with the positional biases of these elements and the sequence preferences across the genome provide strong evidence for their function. Reporter assays would go beyond the scope of this paper.

6) *P.14: "...showing that short transcripts are spliced more slowly...". How do ribosomal proteins behave? They are small, highly expressed, and very often contain introns. Their mRNA expression seems to be optimized at every level. Is splicing an exception to this rule?*

Answer: We have looked into this. We found that ribosomal proteins are slightly more slowly spliced than other ORF-TUs (median of 51 sec. versus 38 sec.). Also the negative correlation between length and splicing is not driven by the ribosome only since it is also found (rank correlation -0.16) when removing ribosome proteins from the analysis. Hence, these observations indicate that splicing rate is not optimized for ribosomal proteins.

7) *Is the increase in splicing rates with transcription rate related to the increase splicing efficiency with expression levels reported on figure 4 of Wilhelm et al (Nature, 2008)?*

Answer: there is a possible confounding when using splicing efficiency as in Wilhelm et al (Nature, 2008). Splicing efficiency is defined as the ratio of reads spanning exon-exon over exon-intron junctions. Steady-state levels of exon-exon junction reads are proportional to synthesis over degradation rates (in our notations μ/λ) whereas the exon-intron junction reads have steady-state values proportional to synthesis over splicing rates (μ/σ). Hence splicing efficiency, which is the ratio of exon-exon over exon-intron reads, equals to splicing rate over

decay rate (σ/λ). Therefore, correlation between expression level (μ/λ) and splicing efficiency (σ/λ) is expected and may solely be explained by variations in degradation rate (λ). The kinetic data allows disentangling these effects.

Junction model, synthesis rate

Junction model, splicing rate

Junction model, decay rate